# Gap junction-mediated glycinergic inhibition ensures precise temporal patterning in vocal behavior

Boris P Chagnaud[1]*, Jonathan T Perelmuter[2], Paul M Forlano[3,4], Andrew H Bass[2]*

[1]Institute of Biology, Karl-Franzens-University Graz, Graz, Austria; [2]Department of Neurobiology and Behavior, Cornell University, Ithaca, NY, United States; [3]Department of Biology, Brooklyn College, City University of New York, Brooklyn, NY, United States; [4]Subprograms in Behavioral and Cognitive Neuroscience, Neuroscience, and Ecology, Evolutionary Biology and Behavior, The Graduate Center, City University of New York, New York, NY, United States

**Abstract** Precise neuronal firing is especially important for behaviors highly dependent on the correct sequencing and timing of muscle activity patterns, such as acoustic signaling. Acoustic signaling is an important communication modality for vertebrates, including many teleost fishes. Toadfishes are well known to exhibit high temporal fidelity in synchronous motoneuron firing within a hindbrain network directly determining the temporal structure of natural calls. Here, we investigated how these motoneurons maintain synchronous activation. We show that pronounced temporal precision in population-level motoneuronal firing depends on gap junction-mediated, glycinergic inhibition that generates a period of reduced probability of motoneuron activation. Super-resolution microscopy confirms glycinergic release sites formed by a subset of adjacent premotoneurons contacting motoneuron somata and dendrites. In aggregate, the evidence supports the hypothesis that gap junction-mediated, glycinergic inhibition provides a timing mechanism for achieving synchrony and temporal precision in the millisecond range for rapid modulation of acoustic waveforms.

*For correspondence:
boris.chagnaud@uni-graz.at (BPC);
ahb3@cornell.edu (AHB)

**Competing interests:** The authors declare that no competing interests exist.

## Introduction

Complex behaviors often depend on temporally precise neuronal firing that coordinates network activity at brain levels ranging from cortical microcircuits to hindbrain pattern generators (*Llinás, 2014*; *Kros et al., 2017*; *Sober et al., 2018*). Mechanisms known to increase precision at single cell and network levels include, for instance, feed-forward inhibition in auditory circuits (*Grothe, 2003*), recurrent inhibitory input in cerebral cortex (*Kapfer et al., 2007*), and neuronal synchrony in cortical and sensory neurons (*Tiesinga and Sejnowski, 2001*; *Uhlhaas et al., 2010*). Synchronous, concurrent activation of neurons is widely distributed in the brain (*Llinás, 2014*) and especially important for behaviors requiring both rapid and precise motoneuron activation such as electrogenesis in fishes (*Bennett, 1971*) and vocalization in fishes (*Chagnaud et al., 2012*) and tetrapods (*Mead et al., 2017*; *Kwong-Brown et al., 2019*).

Several mechanisms by themselves or in combination contribute to neuronal synchrony: coherent excitatory firing, electrotonic coupling, and inhibitory input (*Singer, 1999*; *Uhlhaas and Singer, 2006*; *Kapfer et al., 2007*). While coherent (i.e., phasic) input leads to neuronal coupling mainly by excitation, inhibition might be the predominant way to synchronize activity (*Van Vreeswijk et al., 1994*). Electrotonic coupling enhances synchrony by spreading voltage changes, for example, during synaptic inputs, that lead to concomitant membrane potential changes within an electrotonic, interconnected population (*Bennett and Zukin, 2004*; *Pereda, 2014*).

Unlike motor systems controlling locomotion and respiration (*Ramirez and Baertsch, 2018*; *Grillner and El Manira, 2020*), in-depth inquiries of cellular and network properties of brainstem neurons influencing acoustic signaling remain relatively unexplored, despite advances in characterizing the musculoskeletal periphery (*Mead et al., 2017*; *Kwong-Brown et al., 2019*; *Riede et al., 2019*; *Bowling et al., 2020*). A neurobehavioral challenge often facing soniferous species is fine temporal control of rapid modulations of acoustic waveforms.

The vocal network of toadfishes, a marine order of teleosts that is highly dependent on acoustic signaling for social interactions and successful reproduction, exhibits unusually high levels of synchronous activity, making it ideal for investigating mechanisms underlying precise neuronal firing (reviewed in *Pappas and Bennett, 1966*; *Bass et al., 2015*). This includes Gulf toadfish (*Opsanus beta*), the species studied here, which produce two main types of vocalizations with pulse repetition rates (PRRs) in the range of ~200–250 Hz – broadband agonistic grunts and multiharmonic advertisement calls known as boatwhistles (*Figure 1a*, *Tavolga, 1958*; *Winn, 1967*; *Maruska and Mensinger, 2009*; *Elemans et al., 2014*; *Bass et al., 2015*).

An experimental advantage of the vocal system of toadfishes and other teleosts (*Bass and Baker, 1991*) is that physical attributes of acoustic signals (PRR, fundamental frequency [$F_0$], duration, amplitude modulation) are directly established by a motor volley readily recorded intracranially from hindbrain occipital nerve roots, comparable to hypoglossal roots (*Bass et al., 2008*), in an intact neurophysiological preparation (*Bass and Baker, 1990*; *Bass and Baker, 1991*; *Remage-Healey and Bass, 2006*; *Rubow and Bass, 2009*). These roots give rise to the vocal nerve, which innervates a single pair of so-called 'superfast' muscles attached to the swim bladder that have physiological and molecular properties allowing them to generate power at frequencies 20–100-fold greater than fast and slow twitch muscle, respectively (avian syrinx, bat larynx and rattlesnake shaker also have superfast muscles) (*Rome, 2006*; *Mead et al., 2017*; *Nelson et al., 2018*). To achieve high contraction rates, toadfish superfast muscles require precise neural input provided by the vocal motor volley, termed fictive vocalization in electrophysiological preparations, that is a highly stereotyped, repetitive series of compound nerve potentials (VOC, *Figure 1b*) (motor volleys driving non-superfast muscles, e.g., those used in limb movement, show low temporal coincidence; *McLean et al., 2007*; *Berkowitz, 2008*; *Chagnaud et al., 2012*; *Song et al., 2016*). Individual VOC potentials arise from synchronous motoneuron activity in the midline vocal motor nucleus (VMN) and are matched 1:1 with individual motoneuron action potentials (APs) (*Figure 1c*; *Bass and Baker, 1990*; *Chagnaud et al., 2012*; *Chagnaud and Bass, 2014*) and synchronous contraction of superfast vocal muscles (*Elemans et al., 2014*). Each individual VOC potential represents the synchronous activity of VMN motoneurons; variable potential amplitude reflects different levels of synchronous motoneuron firing and motoneuron recruitment (*Figure 1c*; *Chagnaud et al., 2012*). VOC potential amplitude is easily quantified and serves as a convenient readout of the extent of VMN synchrony (*Chagnaud et al., 2012*). VOCs occur either spontaneously or can be evoked by brief trains of low amplitude, electrical microstimulation in midbrain sites comparable to the periaqueductal gray of birds and mammals (*Kittelberger and Bass, 2013*).

Motoneurons possess 3–5 main dendritic branches and an axon arising from a primary dendrite or soma that lacks collaterals and exits the brain ipsilaterally via the vocal tract (VoTr, *Figure 1d, e*) and joins the ipsilateral vocal nerve root (*Bass and Baker, 1990*; *Chagnaud and Bass, 2014*). Each VMN is bilaterally innervated by adjacent vocal pacemaker neurons (VPN, *Figure 1d, e*) that provide coherent excitatory input and determine VMN firing rate that directly translates into PRR or $F_0$ (see above) (*Bass and Baker, 1990*; *Chagnaud et al., 2011*; *Chagnaud and Bass, 2014*). Both VPN and VMN receive input from a more rostral vocal prepacemaker nucleus (VPP, *Figure 1e*) that encodes call duration (*Chagnaud et al., 2011*; *Chagnaud and Bass, 2014*).

Pappas and Bennett discovered the VMN (*Pappas and Bennett, 1966*) and showed, along with studies of electric fish (*Bennett, 1971*), the contribution of electrotonic coupling to motoneuronal synchrony. This included ultrastructure evidence for gap junction contacts between VMN motoneurons and axons of unidentified origin (likely VPN) (*Pappas and Bennett, 1966*; also see *Bass and Marchaterre, 1989*). More recent studies show that vocal nerve labeling with gap junction impassable tracers only leads to dense retrograde labeling of the ipsilateral VMN, while gap junction passable tracers lead to dense transneuronal, bilateral labeling of VMN, VPN, and VPP (e.g., see *Figure 1d*; *Bass et al., 1994*; *Chagnaud and Bass, 2014*). Besides electrotonic coupling, *Pappas and Bennett, 1966* speculated that inhibitory input to the VMN arose from recurrent

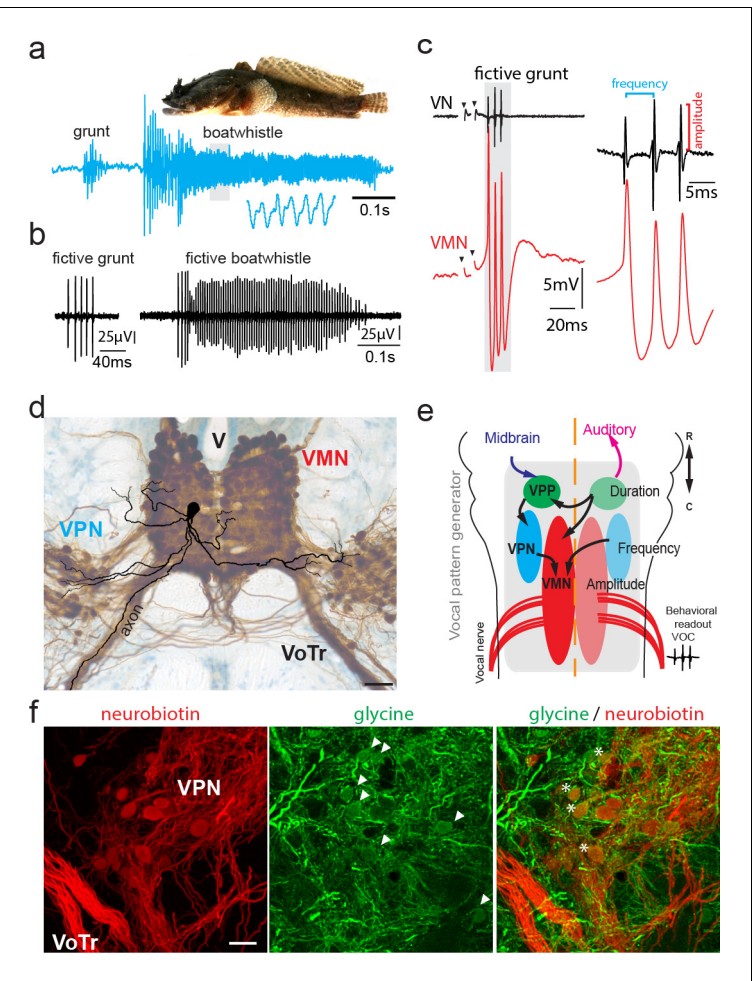

**Figure 1.** Toadfish vocalization and underlying vocal motor circuitry, which includes electrotonically coupled glycinergic neurons. (**a**) Photograph of Gulf toadfish (courtesy of Aaron Rice, Cornell Lab of Ornithology) and waveform of toadfish vocalization (blue) composed of a grunt and a boatwhistle recorded with a hydrophone. Lower waveform is magnified from region outlined by gray box. (**b**) Spontaneous fictive vocalizations (fictive grunt and boatwhistle) recorded from the vocal nerve (VN). (**c**) VN compound potential (fictive call VOC, here a grunt) and intracellular vocal motor nucleus (VMN) motoneuron recording show highly time-locked activity. Right side shows higher magnification and close correlation between motoneuronal action potentials and VN spikes. (**d**) Photograph of transverse hindbrain section showing reconstructed VMN motoneuron superimposed over neurobiotin-filled VMN and vocal pacemaker nucleus (VPN) (dark brown); cresyl violet counterstain. Scale bar in (**d**) represents 100 μm. (**e**) Dorsal schematic view of caudal hindbrain showing toadfish vocal motor circuit comprising three anatomically separate nuclei coding for different attributes: duration coding vocal prepacemaker nucleus (VPP), frequency (pulse repetition rate) coding VPN and amplitude coding VMN. VNs innervate muscles used in sound production. (**f**) Photomicrographs of neurobiotin-labeled (red) VPN neurons and vocal tract (VoTr) (left), glycinergic-immunoreactive (green) neurons (somata indicated by white arrowheads) and fibers (middle), and overlay of both (right). A subset of glycinergic neurons in the VPN are co-labeled with neurobiotin (asterisks); see *Rosner et al., 2018* for detailed quantification. The scale bar is 20 μm.

inhibitory activity (antidromic stimulation of vocal nerve led to hyperpolarization (HYP) at high, but not low, stimulation amplitudes). How this putative inhibition could be induced by activating motoneuron axons in the vocal nerve remained to be tested.

Recent studies of other motor systems provide a framework for *Pappas and Bennett, 1966* observations by showing a role for electrotonic coupling between motoneurons and (inhibitory) pre-motoneurons in patterning vocalization in frogs (*Lawton et al., 2017*) and locomotion in adult zebra-fish (*Song et al., 2016*) and larval flies (*Matsunaga et al., 2017*). We also recently reported in Gulf toadfish a subset of glycinergic VPN neurons transneuronally labeled with the gap junction passable

tracer neurobiotin, strongly suggestive of electrotonic coupling within the vocal network (*Rosner et al., 2018*; see *Figure 1f*). Could the electrotonically coupled glycinergic premotoneurons be involved in the patterning of motoneuron firing in the toadfish vocal pattern generator, as shown for other motor systems?

Here, we took advantage of *Pappas and Bennett, 1966* intracellular recording approach, especially the use of antidromic nerve stimulation to investigate electrotonic coupling, to reveal intrinsic and network properties underlying concurrent motoneuron firing in the VMN. We provide evidence demonstrating coherent, high-frequency excitatory input as well as inhibitory glycinergic input to VMN motoneurons contributing to coordinated network activity. We further show that a strong HYP in these motoneurons after spiking (also see *Pappas and Bennett, 1966*), not seen during intracellular current injection, enhances synchronized vocal network activity and that HYP amplitude directly depends on motoneuron population activation. Next, using pharmacology combined with antidromic stimulation, we provide evidence that glycinergic VPN neurons activated via electrotonic coupling can account for the HYP. In aggregate, these findings strongly suggest that the HYP is mediated by a glycinergic inhibition dependent on gap junctional coupling. Besides its potential involvement in motoneuronal patterning, as recently shown in locomotor systems (see above), this provides an adaptive mechanism to enhance temporal precision in the activation of acoustic signaling networks and perhaps time coding in other motor systems requiring high levels of precision.

## Results

All statistical results are summarized in a table in *supplementary file 1* (see Materials and methods for details on the test used).

### Synchronous motoneuron firing and post-spiking HYP

Midbrain stimulation led to membrane depolarizations in motoneurons that increased in amplitude until a single AP was fired (*Figure 2a1, 2*). This AP coincided with the first appearance of a strong post-spiking HYP and a single VOC nerve potential. With increasing stimulus strength, additional APs were detected that matched 1:1 with additional VOC potentials (*Figure 2a3, 4*). Each repetitive series of VOC potentials mimicked the temporal properties of sound pulses comprising natural grunts (see *Figure 1a*; *Tavolga, 1958*; *Winn, 1967*; *Maruska and Mensinger, 2009*; *Elemans et al., 2014*).

Across repetitions, the amplitude of the second VOC potential always exceeded the first ($73.89 \pm 4.4$ µV vs. $36.6 \pm 4.1$ µV; n = 14 neurons; N = 4 fish; p<0.0001) (*Figure 2a3–5*). The reduced amplitude of the first VOC potential reflected a less synchronous and/or a partial activation of the motoneuron population (*Figure 2a3*). The first and last VOC potentials generally showed activity distributed over a broader time course (*Figure 2b*; also see inset, amplifying end of vocal nerve recording, e.g., of 'weak' synchrony). The amplitude of the first and last VOC potentials thus directly reflected the extent of synchronous motoneuron activation.

Intracellular motoneuron recordings showed broad depolarizations often present during the first and last VMN APs (*Figure 2a3–5, b* ) that often displayed spikelets strongly indicative of asynchronous motoneuron activity (*Figure 2a4, b, c*) (also see *Chagnaud et al., 2012*). The strong variability in amplitude of motoneuron APs and VOC potentials during a given VOC (e.g., *Figure 2a2–5*) was further reflected in different half-widths of the first APs riding on the broad depolarizations. These were significantly wider than those of subsequent APs whose amplitude correlated with a higher amplitude in the corresponding VOC potential (first AP half width: $0.81 \pm 0.11$ ms vs. second $0.62 \pm 0.06$ ms; n = 8, N = 3; p=0.001). As highlighted in *Figure 2b* (but also evident in *Figure 2a4, c*), larger, sharp-peaked VOC potential amplitudes (blue trace) are indicative of synchronous firing across the VMN population and correlated with narrower motoneuron APs (blue trace).

Antidromic activation via the vocal nerve (electrodes implanted in vocal muscles; see Materials and methods) revealed motoneuron APs lacking the prominent HYP at their respective threshold (blue trace, *Figure 2c*, right). This was consistent with intracellular square pulse current injections showing no clear HYP after AP firing (*Figure 2d*). Motoneuron AP and HYP amplitudes (relative to resting membrane potential [RMP]) were significantly larger during VOC activity than following antidromic activation at threshold for AP firing (AP: vocal $29.13 \pm 1.48$ mV vs. antidromic: $21.46 \pm 2.93$

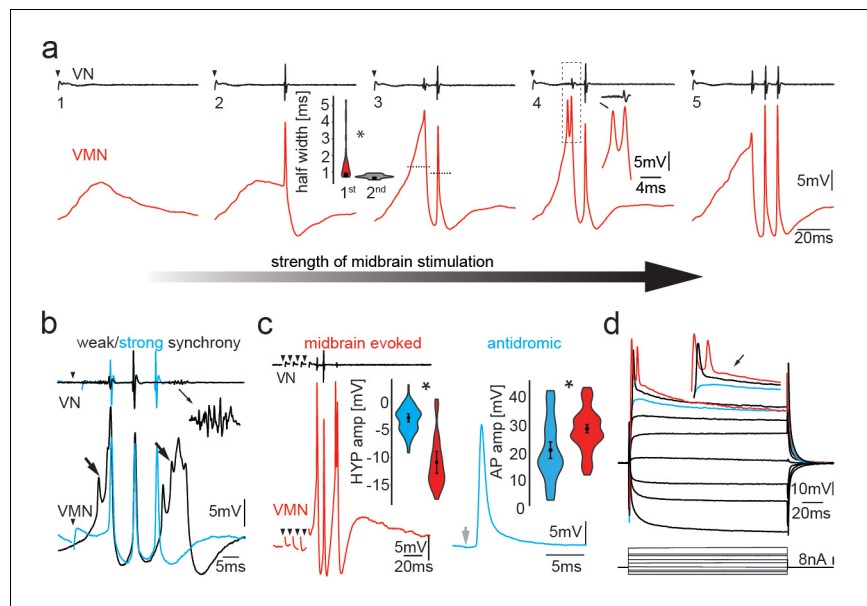

**Figure 2.** Activity in vocal motoneurons reveals prominent differences in action potential shape depending on how action potentials are elicited. (a) Intracellular recordings of a single vocal motor nucleus (VMN) motoneuron showing different stages of activity in the generation of vocal nerve (VN) motor volley, the fictive vocalization, elicited by midbrain electrical stimulation. Violin plot shows the half width (hatched lines in trace 3) of the first and second VMN motoneuron action potentials (APs, for eight neurons). Note the double spikelets that occasionally occur on first APs (inset in 4). (b) Weak (black traces) and strong (blue traces) synchrony of vocal activity is reflected in VN compound potentials (top) and the voltage of intracellularly recorded VMN motoneuron APs (bottom). Electrical artifacts of midbrain stimulation are indicated by black arrowheads. Black arrows indicate spikelets. Inset shows magnification of unsynchronized motor output. (c) Amplitude of VMN motoneuron hyperpolarization (HYP) and AP amplitudes are higher during fictive vocal behavior (red trace) than during antidromic stimulation (blue trace) via the VN. (d) Current voltage responses of a VMN neuron show rapid AP adaptation. Inset shows color-coded traces just before AP initiation (blue), with one (black) and two (red) APs. Note the decrease in second AP height in the red trace. Violin plots show distribution of data from all recordings with estimated means and standard error from the nested mixed model. Asterisks here and in Figures 4, 5, and 7 indicate significant differences.

The online version of this article includes the following source data for figure 2:

**Source data 1.** Comparison between motoneuron action potential and hyperpolarization amplitude during midbrain-evoked vocal and antidromic stimulation.

mV; p=0.024; HYP: vocal −10.92 ± 1.16 mV vs. antidromic −2.95 ± 0.70 mV; n = 12, N = 5: p=0.001, see *Figure 2c*).

## Motoneuron electrical coupling and HYP

To investigate the origin of different motoneuronal AP and HYP amplitudes observed for antidromic-evoked APs and those during VOC activity, motoneurons were stimulated antidromically at varying amplitudes via the ipsilateral vocal nerve root (ad-ipsi, blue traces; *Figure 3*). At low amplitudes, we detected small depolarizations (*Figure 3*) whose amplitude gradually increased with stimulation strength, that is, with increasing recruitment of motoneuron axons and with shapes and peak latencies indicative of electrotonic coupling (3.54 ± 0.28 ms; n = 9, N = 4). Collision experiments using antidromic-activated and intracellular-evoked APs (via intracellular current injection) revealed these APs could not be blocked, that is, they resulted from electrical coupling (*Pappas and Bennett, 1966*; *Kiehn and Tresch, 2002*, *Figure 3—figure supplement 1*).

With increasing amplitude of vocal nerve stimulation, the axon of the respectively recorded motoneuron was eventually recruited and an antidromic AP invaded the recorded motoneuron as shown by the significantly shorter peak latency (1.42 ± 0.31 ms after stimulation; n = 11, N = 4) compared to the previously mentioned subthreshold depolarization (p=0.003) (*Figure 3*). These APs showed

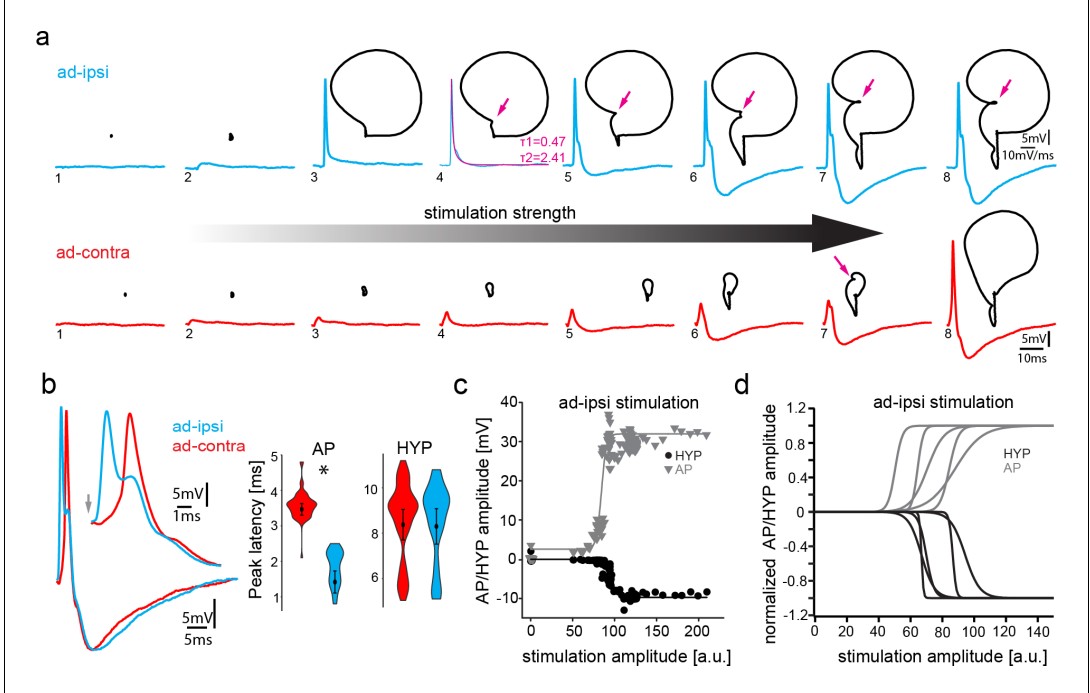

**Figure 3.** Antidromic stimulation activates motoneuronal hyperpolarization (HYP) that depends on stimulation amplitude. (a) Intracellular record of antidromically activated motoneuron upon ipsi- (blue) and contralateral (red) activation with increasing stimulation amplitude (schematized by big horizontal arrow). Black lines show phase plane plots, and magenta arrows indicate additional depolarization prior to HYP onset. Magenta line in fourth trial represents exponential fit for this recording, with the constants indicated. Scale bars on top row for phase plane plots, bottom row for color traces. (b) Overlay of ipsi- and contralateral stimulation of neuron shown in (a). Violin plots show HYP and action potential (AP) peak latencies. (c) AP (gray) and HYP (black) peak amplitude of VMN neuron response upon ipsilateral antidromic stimulation of variable amplitude (arbitrary units) and corresponding sigmoid fits (color coded lines). (d) Normalized sigmoid fits of different neurons showing differences in recruitment threshold by antidromic stimulation, but similar time courses.

The online version of this article includes the following source data and figure supplement(s) for figure 3:

**Source data 1.** Comparison between motoneuron action potential and hyperpolarization latency during midbrain-evoked vocal and antidromic stimulation.

**Figure supplement 1.** Collision experiments reveal gap junctional coupling in vocal motoneurons.

**Figure supplement 2.** Potential reafferent input via the dorsal roots does not contribute to the motoneuronal hyperpolarization (HYP).

**Figure supplement 2—source data 1.** Comparison between motoneuron action potential and hyperpolarization amplitude during baseline conditions and after cutting the dorsal roots.

**Figure supplement 3.** Motoneurons are able to repetitively fire under a pulse train condition, but fail in response to a permanent current injection, revealing the necessity of a hyperpolarization for correct vocal patterning.

no HYP, but instead were characterized by a slow decay (double exponential fit; average time constant $\tau1$: $0.89 \pm 0.25$; time constant $\tau2$: $7.76 \pm 2.48$; n = 5, N = 2) back to the RMP (example shown in magenta trace in *Figure 3a*). Surprisingly, an HYP started to appear with increasing antidromic stimulation amplitude (*Figure 3*). The AP and HYP peak amplitudes increased with stimulation strength until each reached a plateau (*Figure 3a*). A HYP was never observed during intracellular square pulse current injections (*Figure 2d*), raising the question on the origin of this HYP.

Phase plane plots of the membrane potential during ipsilateral antidromic activation revealed further changes in motoneuronal activity upon increasing stimulation amplitudes (*Figure 3a*, black traces). The gradual appearance of the HYP, together with the absence of the HYP upon initial AP firing ((*Figure 3a*)) suggested that a further recruitment of motoneurons via the antidromic stimulation underlies HYP generation ((*Figure 3a*)). Phase plane plots further revealed an additional component: a broadening of the depolarization after AP firing (magenta arrows, (*Figure 3a*)). As this depolarizing component was not present at the recruitment threshold, it cannot have originated

from the gap junction-mediated coupling superimposed on the AP. In a few cases, this depolarizing component eventually led to a second AP firing (not shown).

A Renshaw cell-like recurrent inhibition in which spinal motoneurons use an axon collateral to activate a local inhibitory circuit (*Renshaw, 1941*; *Eccles et al., 1954*), comparable to the 'apparently recurrent inhibition' proposed by *Pappas and Bennett, 1966* to account for inhibitory input to the VMN, could be ruled out as the origin of the HYP given the lack of motoneuron axon collaterals (*Figure 1d*, *Chagnaud and Bass, 2014*) and the rapid onset of the HYP. To exclude that a motoneuronal axon collateral could arise at the periphery and enter via one of the nearby dorsal roots, the dorsal roots were bilaterally cut in two experiments (*Pappas and Bennett, 1966* used this method to show that antidromically evoked APs in motoneurons did not arise from afferents). There was no difference in HYP amplitude (% of baseline) between cut and uncut recordings during VOCs (before cut: −8.21 ± 3.1 mV vs. after: −7.51 ± 1.22 mV; n = 18, N = 2: p=0.8) (*Figure 3—figure supplement 2*) or during antidromic stimulation (before cut: −3.2 ± 0.59 mV vs. after: −4.84 ± 0.71 mV, n = 18, N = 2: p=0.117). These results excluded a motoneuronal collateral via one of the dorsal roots as the origin of the HYP.

Contralateral antidromic stimulation also revealed electrotonic potentials whose amplitude depended on stimulation strength (ad-contra, red traces; *Figure 3a*). Electrotonically mediated potentials eventually reached threshold and evoked an AP. The peak latency of these APs was significantly longer (3.47 ± 0.16 ms; n = 11, N = 4; p=0.004) than ones elicited ipsilaterally (*Figure 3b*), while the peak latency of the ipsilaterally evoked subthreshold depolarization did not differ, consistent with their common origin from electrotonic coupling (see above). As tract tracing and intracellular neuron fills showed that motoneurons only innervate the ipsilateral muscle (*Chagnaud and Bass, 2014*), electrical coupling alone is thus able to drive AP firing, independent of whether motoneurons belong to the ipsilateral or contralateral VMN population.

As with the ipsilateral antidromic activation, an HYP component could clearly be distinguished in the contralateral antidromic stimulation experiments (*Figure 4a*). This HYP occurred independent of AP firing, emphasizing the independence of the two events within a given motoneuron.

Consistent with our findings in the closely related midshipman toadfish, *Porichthys notatus* (*Chagnaud et al., 2012*), the ability to initiate an AP via electrotonic coupling was in strong contrast to our intracellular current injections that failed to initiate an AP in most cases, even at high current intensities (>5 nA). In cases where intracellular current injection elicited an AP, motoneurons showed rapid adaptation of AP firing, likely due to weak somatic repolarization ability (see *Figure 2d*). Gulf toadfish are, however, able to repetitively contract their superfast vocal muscles for several hundred milliseconds (*Figure 1a*). How can the muscle achieve this if motoneurons cannot fire for extended time periods due to the rapid AP adaptation seen during square pulse current injections? To test whether the HYP was required to de-inactivate motoneurons, we stimulated the motoneurons in which current injection led to AP firing with pulse trains of different frequencies. In contrast to long (>50 ms) duration pulses (*Figure 2d*), motoneurons showed no signs of AP adaptation to pulse trains with brief (<5 ms) pulses, indicating the necessity of membrane repolarization for sustained motoneuron firing (*Figure 3—figure supplement 3*). Stimulation was reliable into the behaviorally relevant physiological range (the PRR/fundamental frequency of toadfish vocalizations) with train frequencies tested up to 110 Hz. The weak repolarization capability and low excitability of the motoneurons thus provide the means to prevent sustained AP firing, which would decrease the extent of firing synchrony and precision across the VMN population.

## Network activity induces HYP

The presence of the HYP only at high antidromic stimulation amplitudes, that is, high levels of motoneuron recruitment, strongly suggested a network-dependent activation of the HYP. To test this hypothesis, we ipsilaterally evoked an AP antidromically in a VMN motoneuron (ad-ipsi) at low threshold stimulation (i.e., without an HYP), followed by stimulation of the contralateral nerve (ad-contra), which resulted in a small electrotonic depolarization (*Figure 4a, b*). Subsequently, the delay of this second stimulation was reduced up to the time point of the first ipsilateral nerve stimulation (*Figure 4b*). Once close to the antidromic AP, an HYP started to appear that increased in amplitude the closer the contralateral-evoked potential came to the ipsilateral-evoked antidromic AP (*Figure 4b*; heat map in *Figure 4a*; also see *Video 1*). These experiments suggested that an increase in overall depolarization in the vocal network is needed to generate the HYP. To test this hypothesis,

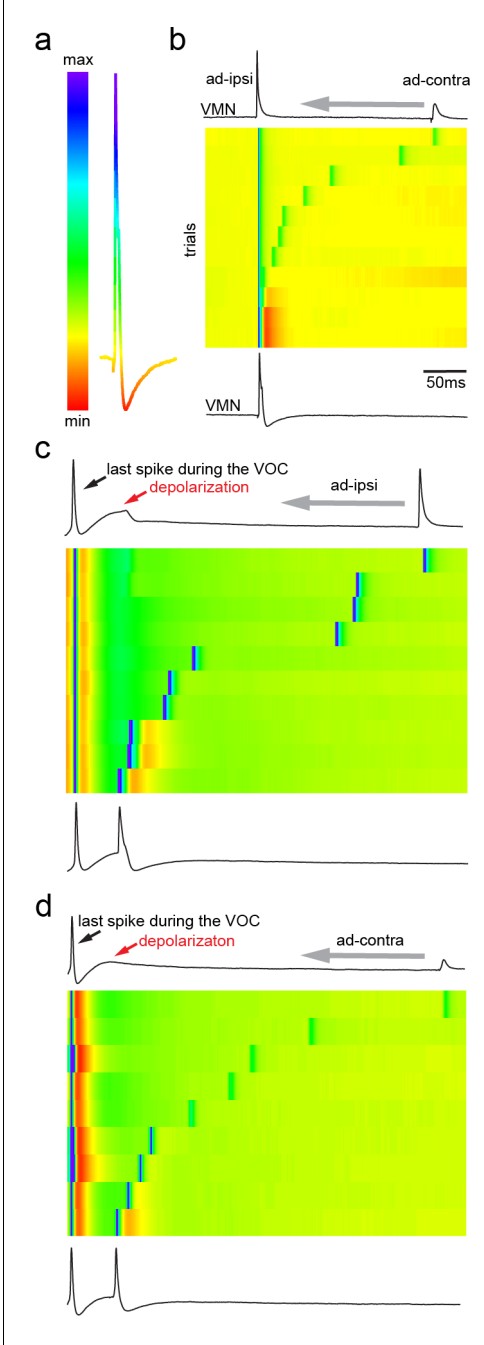

**Figure 4.** Antidromic activation of vocal motor nucleus (VMN) motoneurons revealing network-dependent hyperpolarization (HYP). (a) Antidromic-evoked action potential (AP) color-coded to relative voltage amplitude and corresponding color bar. (b) Stimulation-dependent voltage matrix (SDVM; color code given in [a]) of ipsilateral antidromic activation followed by contralateral antidromic activation (that generated only a depolarizing potential) of decreasing latency revealing the appearance of a HYP when ipsilateral and contralateral stimulation overlap. In this situation, ipsilateral antidromic stimulation was set to elicit an AP without a HYP. With decreasing distance of

we took advantage of the prominent, wide depolarization that often appeared in motoneurons at the end of a VOC (see *Figures 1c* and *2c*). Similar to the above, we moved an ipsi- or contralateral, antidromically evoked depolarizing potential into this depolarization occurring at the end of a VOC (ad-ipsi and ad-contra, *Figure 4c, d*, respectively). With decreasing lag between the antidromically evoked potential and the depolarization during the VOC, an HYP started to appear in the contralateral-evoked potential that increased in amplitude (*Figure 4c, d*). This again showed the necessity of a network-wide depolarization in order to elicit the HYP.

## Single motoneuron activity and HYP reflect network synchrony

To test the contribution of single motoneurons to network activity, we performed intracellular recordings of motoneurons using QX314 that blocks voltage-dependent sodium channels intracellularly (*Yeh, 1978*). After intracellular iontophoresis of QX314, motoneurons exhibited a small, but significant, decrease in AP amplitude compared to baseline conditions during VOCs (*Figure 5a*) (baseline: $31 \pm 2.67$ mV vs. QX314: $27.27 \pm 2.53$ mV; n = 7, N = 2; p=0.015). There was a much more prominent decrease in antidromically evoked AP amplitude relative to baseline (baseline: $21.92 \pm 7.16$ mV vs. QX314: $8.90 \pm 6.82$ mV; n = 7, N = 2; p=0.006) (*Figure 5b*). While a significant difference in the HYP during VOCs was observed (baseline: $-9.87 \pm 0.75$ mV vs. QX314: $-2.19 \pm 0.95$ mV; n = 7, N = 2; p=0.003), no significant change could be detected in the HYP following antidromic activation (baseline: $-4.44 \pm 3.57$ mV vs. QX314: $-4.3 \pm 3.56$ mV, n = 7, N = 2; p=0.703) (*Figure 5a, b*). This seemingly contradictory result is likely due to the electrotonic coupling of the network where other motoneurons contribute to the potential of individual motoneurons. Even though unlikely, due to the effect on the antidromic AP, an alternative interpretation is that an insufficient quantity of QX 314 was injected. Manipulation of only one out of the hundreds of motoneurons in VMN via QX314 injections is thus not sufficient to reveal the full extent of the HYP activity. These experiments demonstrated that (i) the contribution of the recorded motoneuronal AP firing to the firing of that motoneuron is rather small during VOC activity (the activity of the neuron is dominated by gap junction-coupled potentials); (ii) during a VOC, the HYP amplitude of the recorded neuron only partly depends on its firing an AP (also see above); and (iii) most of

*Figure 4 continued*

the contralateral stimulation, a HYP started to appear (inset in SDVM). Top and bottom black lines represent first and last rows of the SDVM in this and in the following panels. (c) SDVM showing last compound potential of a fictive vocalization (VOC) with associated depolarization (depol) and antidromic stimulation (set to elicit only a depolarization) of ipsilateral vocal nerve with decreasing latency. Ipsialateral potential generated an AP with decreasing distance to the depolarization that was accompanied by a HYP with further decrease in latency. (d) As in (c) but with contralateral antidromic stimulation.

the activity displayed by a given motoneuron reflects population-level motoneuronal activity.

## Necessity of gap junctional coupling for HYP activation

Having observed that the HYP is highly dependent on activation of the VMN population and not on single-neuron AP firing (*Figures 3–5*), we next blocked gap junctional coupling to determine if the HYP originated from a network activation. A combined superfusion of the exposed vocal hindbrain region coupled with pressure injection of carbenoxolone (CBX, a gap junction blocker) directly into VMN severely impaired the vocal network's ability to generate synchronized motor discharges as evidenced by barely detectable VOC activity (*Figure 5c*, red trace). However, even very low-amplitude VOC-related activity could still be detected in intracellular recordings from motoneurons upon midbrain stimulation, showing that the vocal network could still be activated. Upon antidromic stimulation (*Figure 5d*), APs had a similar amplitude as they did in controls (see violin plots in *Figure 5d*; baseline: 17.80 ± 2.05 mV vs. CBX: 16.15 ± 2.14 mV; n = 25, N = 3; p=0.59), showing that the loss of vocal-related activity during midbrain activation was not due to CBX impairment of motoneuron AP-generating capacity, but to decreased electrotonic input to the motoneurons. No HYP could be elicited in antidromic-activated motoneurons (baseline: −4.79 ± 0.94 mV vs. CBX: −0.79 ± 0.97 mV; n = 25, N = 3; p<0.0001). This showed that gap junctional coupling is indeed required to activate the HYP.

## Dependence of HYP activation on glycinergic inhibitory input to VMN

As noted in the Introduction, we previously identified a subpopulation of glycinergic VPN neurons, suggesting direct glycinergic input onto motoneurons (also see *Figure 1f*). To confirm and identify the location of glycine release onto vocal motoneurons, we used multicolor super-resolution structured illumination microscopy (SR-SIM) imaging, which can achieve an enhanced resolution of ~100 nm in the lateral (x–y) and ~300 nm in the axial (z) planes (*Gustafsson, 2008*). SR-SIM imaging demonstrated dense, punctate glycinergic-immunoreactive labeling throughout VMN. Overlap of glycinergic signal with labeling for synaptic vesicle protein (SV2) revealed prominent boutons directly abutting motoneuron somata (*Figure 6a–c*). Glycinergic boutons were also observed on motoneuron dendrites within the contralateral VMN (*Figure 6d–h*) and VPN (*Figure 6d, i–k*). These results demonstrated an anatomical basis for glycinergic release and inhibition spatially distributed across the entire somato-dendritic extent of vocal motoneurons.

Having identified glycinergic contacts on the VMN motoneurons and that blocking gap junctional coupling was essential for the HYP (see above), we next carried out a series of experiments to more directly investigate if inhibition was responsible for the HYP (*Figure 7a–c*). We observed early on that at midbrain stimulation levels sub-threshold to VOC compound potential induction, tonic membrane HYPs (on average −2.48 ± 0.68 mV below RMP; n = 7; N = 2) could be detected, indicating activation of inhibitory

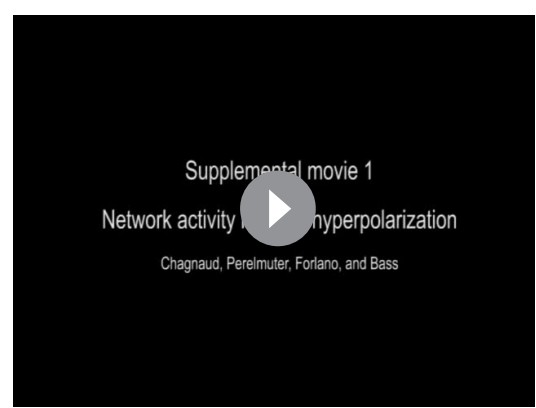

**Video 1.** Intracellular recording of a vocal motoneuron during sequences of repeated antidromic stimulation via the vocal nerve on the ipsilateral and contralateral sides (both subthreshold to network activation of glycinergic input). The latency between the ipsi- and contralateral stimulation was diminished, which eventually led to the generation of a hyperpolarization via glycinergic input.
https://elifesciences.org/articles/59390#video1

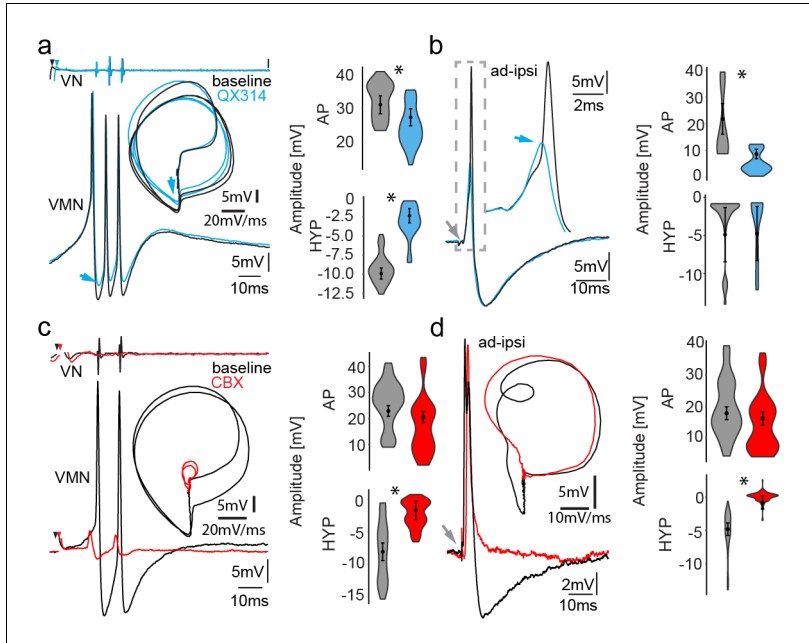

**Figure 5.** Network activity and gap junctional coupling are essential to generate the motoneuron hyperpolarization (HYP). (**a, b**) Baseline (black) intracellular recording of vocal motor nucleus (VMN) neuron action potentials (APs) and with QX314-filled electrodes (blue) during fictive vocal activity (top trace vocal nerve, VN) (**a**) and ipsilateral antidromic stimulation (**b**). Inset in (**a**) shows phase plane plot and in (**b**) higher magnification of antidromically evoked APs. Violin plots show percent change from baseline level for AP and HYP during either condition. Blue arrows point to the differences in AP and HYP levels. (**c, d**) Intracellular recording of one VMN neuron before (black) and one VMN neuron after (red) blocking gap junctions by application of carbenoxolone (CBX). Inset shows phase plane plot. Violin plots show amplitude decrease in baseline for AP and HYP during either condition. Plot shape shows distribution of data from all recordings. Center dot and error bars show estimated means and standard error derived from the nested mixed model. Gray arrow in(b) and (d) indicates stimulus onset.

The online version of this article includes the following source data for figure 5:

**Source data 1.** Comparison between motoneuron action potential and hyperpolarization amplitude during mid-brain-evoked vocal and antidromic stimulation for QX314 treatment and carbenoxolone treatment.

inputs (*Figure 7—figure supplement 1a*, left; magenta trace: subthreshold, black trace: suprathreshold); these HYPs were not artifacts originating from electrical stimulation (*Figure 7—figure supplement 1a, r*ight, orange trace). Due to the high synchrony of motoneuron APs during vocal activity, field potentials could be detected even with our high-resistance electrodes (*Figure 7—figure supplement 1a*, right, black arrow). We also found that changing the chloride reversal potential by intracellular chloride injections via 3 M KCl-filled electrodes revealed a prominent inhibitory input to motoneurons during VOCs as the membrane potential showed significant changes in the degree of repolarization compared to baseline levels (*Figure 7—figure supplement 1b*, blue arrow). The HYP during VOCs was heavily reduced (baseline: −9.62 ± 1.34 mV vs. chloride injected: −1.20 ± 1.23 mV; n = 10, N = 3; p<0.001) (*Figure 7—figure supplement 1b*, blue trace), thus indicating an inhibitory contribution to vocal behavior. Antidromic stimulation still showed the HYP, however, at a reduced amplitude (baseline: −5.28 ± 1.95 mV vs. chloride injected: −3.30 ± 1.71 mV; n = 9, N = 3; p=0.093) (*Figure 7—figure supplement 1c*). As with QX314 intracellular injections, this seemingly contrary result is likely due to electrotonic coupling whereby manipulating one of hundreds of motoneurons does not reveal the full extent of inhibitory activity in the VMN.

To test the influence of glycinergic contacts onto motoneurons, we pressure injected the glycine receptor antagonist strychnine into the VMN. After strychnine injection, no change in AP amplitude during midbrain-evoked VOCs (*Figure 7a*) could be detected. Strychnine, however, abolished the HYP of motoneurons during VOCs (baseline −4.67 ± 1.88 mV vs. strychnine: 0.89 ± 1.88 mV, n = 20, N = 2; p<0.001) (*Figure 7a*). The HYP during antidromic stimulation completely disappeared

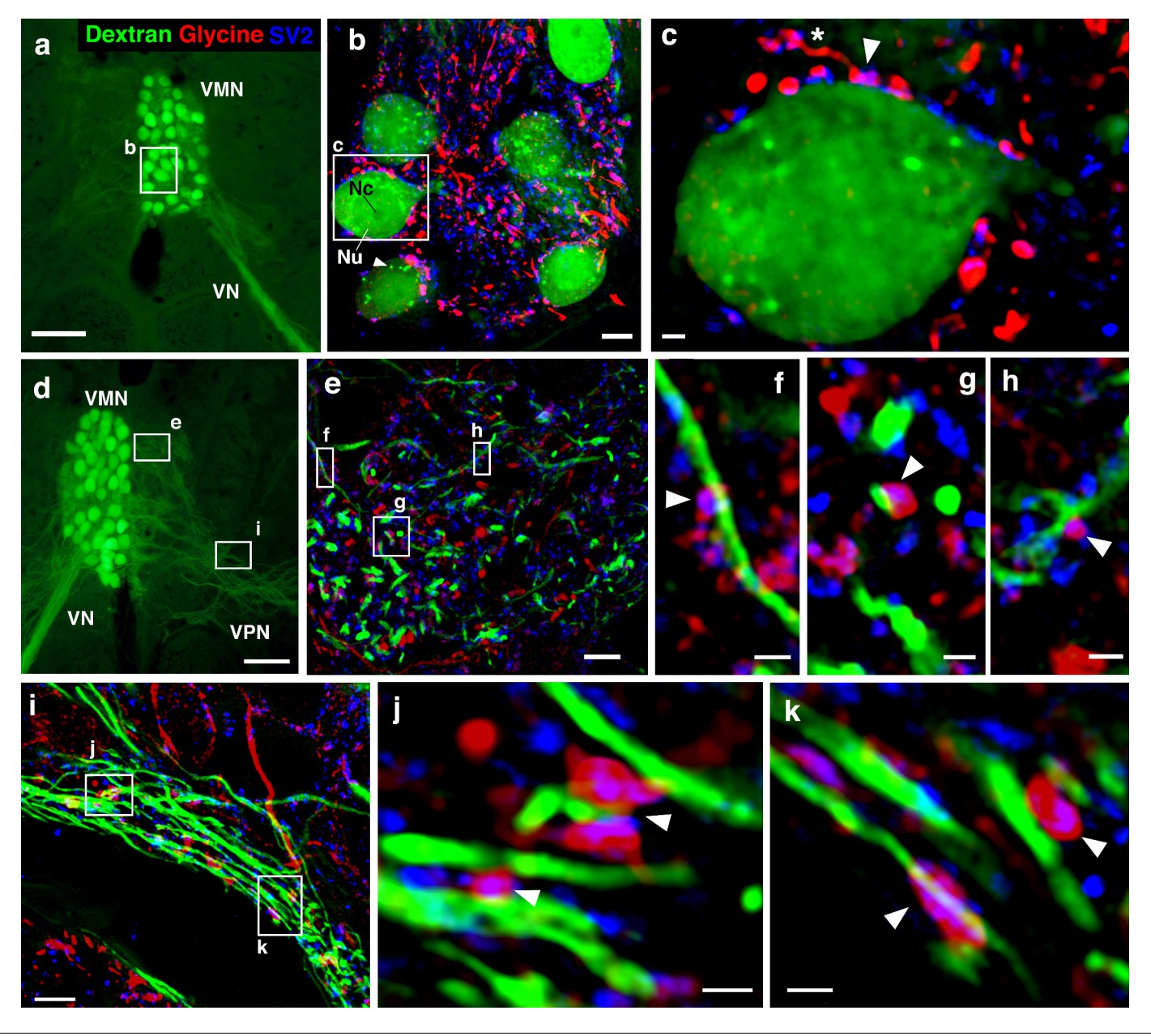

**Figure 6.** Glycinergic input on motoneuron somata and dendrites reveals an anatomical substrate for the motoneuronal hyperpolarization.  (a, d) Widefield micrographs show dextran-filled motoneurons (green) in the vocal motor nucleus (VMN) of two different fish. Dextran labeling of one vocal nerve (VN) results in filling of ipsilateral VMN somata and dendrites that extend into the contralateral VMN (d, box e) and bilateral into adjacent VPN columns (d, box i; also see *Figure 1c*). White boxes correspond to maximum projections of super-resolution structured illumination microscopy (SR-SIM) z-stacks (b, e, i) that show glycine (red) and synaptic vesicle protein 2 (SV2, blue) immunoreactive (-ir) label in proximity to VMN somata (b) and dendrites (e, i). (b) VMN somata are surrounded by dense glycinergic and SV2-ir puncta. A large nucleus (Nu) with nucleolus (Nc) is evident in several somata; punctate dextran labeling within the cytoplasm (white arrowhead) likely indicates sequestration of dextran into vesicles. (c) A single optical section (0.1 μm) from SR-SIM z-stack in (b) shows a glycinergic-ir fiber (white asterisk) forming a bouton (white arrowhead) on a motoneuron soma. Overlap of glycine and SV2-ir (magenta, white arrowhead) in the bouton indicates a site of neurotransmitter release and a likely synapse. Additional optical sections show glycinergic release sites (white arrowheads) on motoneuron dendrites extending into the contralateral VMN (f, g, h, corresponding to boxes in e) and bilaterally into VPN columns (j, k, corresponding to boxes in i). Scale bars represent 100 μm in widefield micrographs (a, d), 5 μm in SR-SIM maximum projections (b, e, i) ,and 1 μm in SR-SIM optical sections (c, f, g, h, j, k).

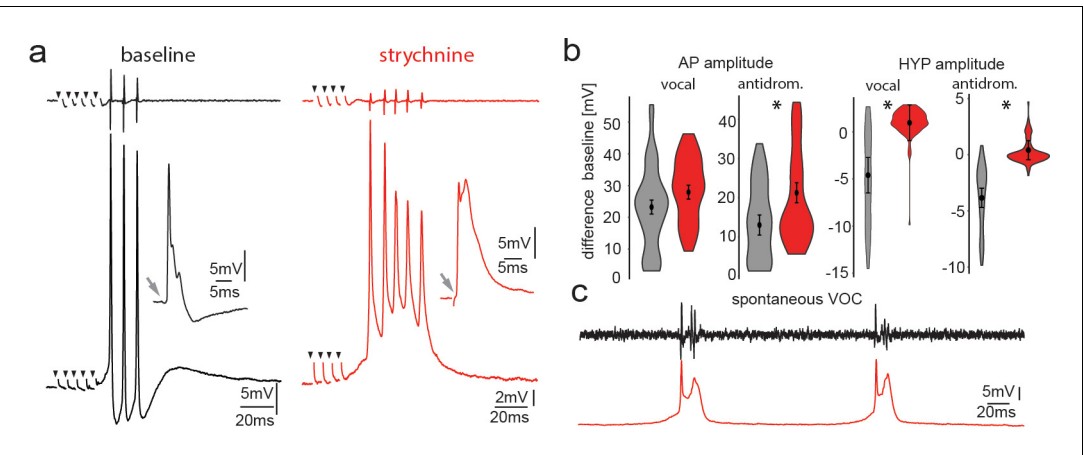

**Figure 7.** Pharmacological blocking of glycinergic input with strychnine shows its importance in vocal patterning. (a) Intracellular recordings from a motoneuron before (black) and another motoneuron after (red) strychnine injection into vocal motor nucleus (top trace vocal nerve). Insets show antidromic-evoked action potentials for the respective neurons. (b) Violin plots showing the change in amplitude of the action potential (AP) and the hyperpolarization (HYP) for control (gray) and strychnine injected (red). (c) Intracellular recordings of a motoneuron (red trace) during two spontaneous fictive vocalizations (VOC, black trace).
The online version of this article includes the following source data and figure supplement(s) for figure 7:

**Source data 1.** Comparison between motoneuron action potential and hyperpolarization amplitude during mid-brain-evoked vocal and antidromic stimulation for strychnine treatment.
**Figure supplement 1.** Inhibitory input is present in the motoneuronal network.
**Figure supplement 1—source data 1.** Motoneuron hyperpolarization amplitude at subthreshold to vocal activity and before and after chloride injection.

(baseline: $-3.85 \pm 0.86$ mV vs. strychnine: $0.40 \pm 0.86$ mV, n = 20, N = 2; p<0.001) (*Figure 7a*, insets). These results suggested that glycinergic neurons, activated via gap junctional coupling, were responsible for generating the HYP during antidromic stimulation.

Following strychnine injections, spontaneous VOCs appeared with similar decreases in motoneuron HYP amplitude not seen for spontaneous VOCs under control conditions without prior strychnine injections (*Figure 7c*). Since VMN motoneurons lack axon collaterals (*Figure 1d*), the results imply that gap junction coupling between motor and glycinergic VPN neurons is sufficient to drive AP firing of glycinergic neurons.

The interval between the first and second motoneuron AP significantly decreased following strychnine application (baseline: $9.61 \pm 0.69$ ms, strychnine $7.94 \pm 0.69$ ms; n = 10, N = 2; p=0.049). Variability of the interspike interval (ISI), as measured by the coefficient of variation, also increased significantly (baseline: $4.90 \pm 3.95\%$, strychnine: $13.54 \pm 3.95\%$; n = 10, N = 2; p=0.009). Thus, strychnine increases both frequency and variability of motoneuron firing that matches VOC output.

Typically, VOCs show sharp peaks between successive potentials, similar to sound pulses within natural grunts (e.g., *Maruska and Mensinger, 2009*; *McIver et al., 2014*). Spontaneous VOCs following injection with strychnine had potentials varying in width (*Figure 7c*) and multiple peaks, reminiscent of VOCs associated with weak VMN synchronization (*Figure 2b*).

## Vocal premotoneurons are excited by gap junctional coupling

*Pappas and Bennett, 1966* also recorded from 'prejunctional fibers' using both orthodromic and antidromic stimulation, although they did not know the origin of these fibers. We have since identified these as VPN neurons (*Bass and Baker, 1990*; *Chagnaud et al., 2011*; *Chagnaud and Bass, 2014*). The pattern of VPN activity directly predicts the pattern of VMN activity, the intervals of VOC potentials, and vocalization PRR (*Figure 8a*, *Chagnaud et al., 2012*). Antidromic stimulation of the vocal nerve leads to membrane depolarizations (via electrotonic input) in premotor VPN neurons in the closely related midshipman (*Bass and Baker, 1990*). We recorded from toadfish VPN neurons to test whether gap junctional coupling is in fact able to induce AP firing in the VPN population, which

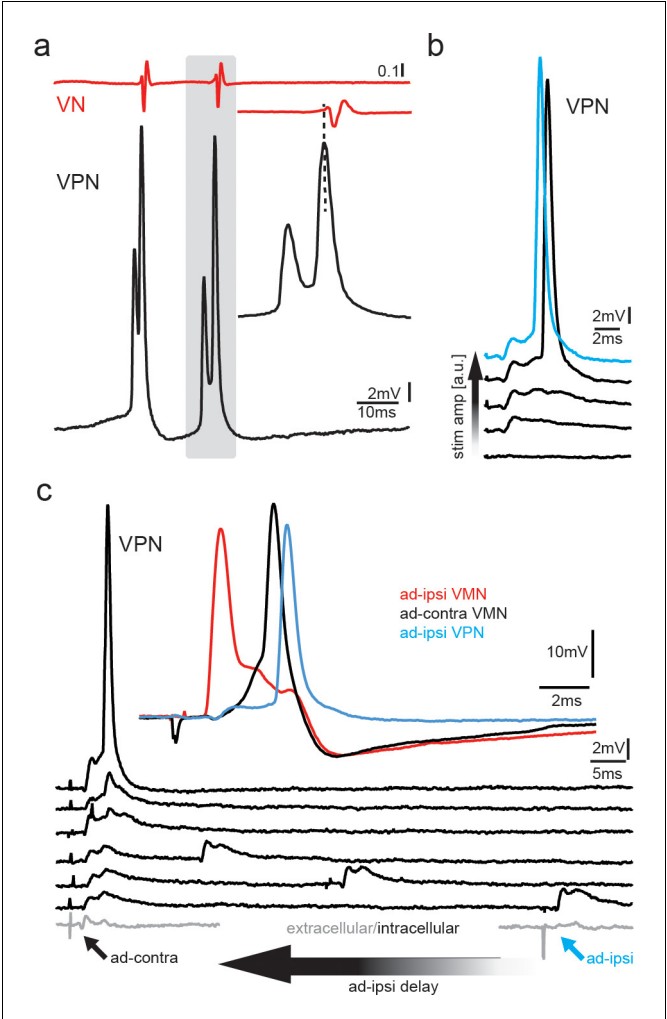

**Figure 8.** Premotoneurons can be activated via gap junctional coupling, thus revealing their potential to contribute to the motoneuron hyperpolarization. (**a**) Intracellular recording of vocal pacemaker nucleus (VPN) neuron during fictive vocal activity shows the characteristic double spiking of VPN neurons. Inset shows timing between vocal nerve (VN) and VPN neuron activity. (**b**) Waterfall plot of intracellular VPN recordings during ipsilateral antidromic vocal nerve stimulation leads to depolarizing potentials in VPN neurons that eventually lead to action potential (AP) firing with increasing stimulation amplitude. (**c**) Waterfall plot of intracellular recordings from a VPN neuron during antidromic stimulation of the ipsi (ad-ipsi) and contralateral (ad-contra) vocal nerve. With decreasing latency between the ad-contra and ad-ipsi stimulation (at identical stimulation amplitudes), an AP can be elicited, showing the capability of gap junctional coupling to induce AP firing. Inset shows recordings from a VMN neuron during antidromic stimulation of the contralateral and ipsilateral sides and from a VPN neuron during ipsilateral antidromic stimulation (not simultaneously recorded).

would be needed to activate or inhibit motoneurons. Toadfish VPN neurons generate two depolarizing components, one before and one during VOCs, that might reflect motoneuron activity leaking through gap junctions (*Figure 8a*, *Chagnaud and Bass, 2014*). During antidromic stimulation of the vocal nerve, VPN neurons showed small subthreshold potentials similar to VMN neurons indicative of gap junctional coupling, as well as APs at higher stimulation amplitudes (*Figure 8b*). Thus, antidromic activation in motoneurons could elicit AP firing in VPN premotoneurons. The latency of these APs (ipsi: 4.66 ± 0.01 ms, n = 3; N = 1; contra: 5.43 ± 0.16 ms; n = 3; N = 1) roughly coincided with the depolarization following antidromically elicited APs in motoneurons (*Figure 3a*, magenta arrow).

To test if a network component might also be important to activate VPN neurons, we stimulated the ipsilateral and contralateral vocal nerve (at varying latencies) to antidromically generate subthreshold membrane depolarizations in VPN neurons. A decrease in latency between the ipsilateral

and contralateral stimulation pulses eventually led to AP firing in VPN neurons, thus showing that AP firing in VPN neurons also depends on gap junctional activation of the vocal network (*Figure 8c*). VPN activation could thus have led to motoneuron depolarization (*Figure 3*, magenta arrows) and activation of gap junction-coupled glycinergic neurons (*Rosner et al., 2018*) that, in turn, inhibit motoneurons and generated the HYP.

## Discussion

Together, our experiments support the hypothesis that gap junction-mediated activation of glycinergic neurons can account for remarkable synchrony in motoneuron firing and likely also contributes to pattern generation. The level of synchronicity in the VMN network is perhaps rivaled only by that of fish electromotor systems where electrotonic coupling and intrinsic neuronal properties seem to play a predominant role (*Bennett, 1971*; *Moortgat et al., 1998*; *Moortgat et al., 2000a*; *Moortgat et al., 2000b*).

Frogs, birds, marine mammals, and humans typically come to mind when we think about sound-producing vertebrate species (*Chen and Wiens, 2020*). Yet, it is also widespread among fishes. Bony vertebrates include the Sarcopterygii, the vast majority of which are tetrapods, and Actinopterygii, which represent more than half of living vertebrate species, close to 99% of which are teleosts, including toadfishes (*Nelson et al., 2016*). Teleost families with evidence for soniferous behavior contain nearly two-thirds of actinopterygian species (*Rice et al., 2020*). Concurrent activation of neurons required for rapid and precise activation of muscle groups underlying acoustic signaling in different lineages of fishes (*Chagnaud et al., 2012*; *Kéver et al., 2020*) and in tetrapods (*Mead et al., 2017*; *Kwong-Brown et al., 2019*) might all benefit from gap junction-mediated glycinergic inhibition.

### Electrical coupling and inhibition in a vocal network

Electrotonic coupling is known to increase the synchronous activity of neuronal populations (*Marder et al., 2017*; *Alcamí and Pereda, 2019*). An important and novel role of gap junctional coupling in the toadfish vocal network is its involvement in generating the prominent HYP in toadfish motoneurons detected during both midbrain-evoked VOCs and antidromic motoneuron stimulation. This HYP, which could not be elicited upon intracellular current injection, is essential to generating vocalizations with high temporal precision.

A distinguishing temporal feature between grunts and boatwhistles, the two main call types of toadfishes, is the greater stability of time intervals between successive sound pulses in boatwhistles (*Maruska and Mensinger, 2009*), which is also the case for the grunt and boatwhistle-like 'hum' of midshipman (*Brantley and Bass, 1994*; *McIver et al., 2014*). Our results strongly imply a salient role for glycinergic premotoneurons in determining this temporal feature of toadfish calls that varies with social context (aggression versus courtship). Interestingly, for fish of both sexes collected during the non-breeding season, the HYP did not appear during antidromic stimulation, although it was present during the VOC (not shown). While we did not quantify this result, a seasonal variability in the expression of the HYP upon antidromic

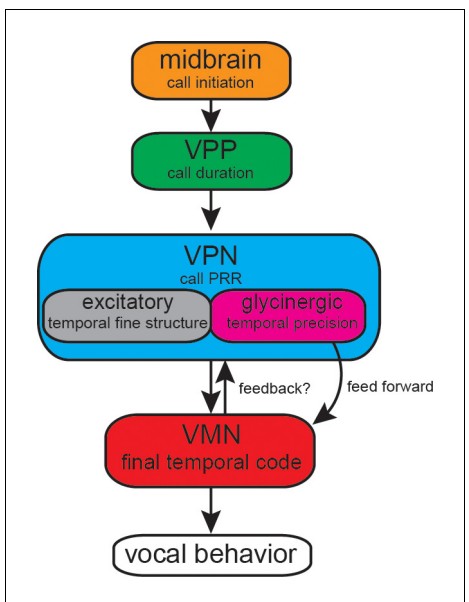

**Figure 9.** Summary of proposed action of glycinergic vocal pacemaker nucleus (VPN) neurons in the activation sequence generating vocal behavior. Temporal activation of the hyperpolarization strongly suggests a feed-forward inhibitory role. Antidromic experiments indicate the possibility of a feedback mechanism that might affect temporal patterning as seen in other motor systems. VPP: vocal prepacemaker nucleus; VMN: vocal motor nucleus; PRR: pulse repetition rate.

stimulation appeared to be present. This variation might be due to the previously observed seasonal variability in expression levels of connexin transcripts in the closely related midshipman fish (*Feng et al., 2015*).

## Feedback or feed-forward inhibition?

Our antidromic stimulation experiment suggests a feedback loop of glycinergic inhibitory neurons activated upon by depolarization of the VMN–VPN network. Antidromic activation is a highly artificial condition in which depolarizations are first elicited in VMN, and then transmitted via gap junctions to VPN. Thus, a recurrent inhibitory pathway is activated during antidromic stimulation. However, the shorter latency of the inhibitory potential during vocalization compared to antidromic activation argues against a recurrent inhibition hypothesis and favors a feed-forward hypothesis. During natural vocal behavior, VPN would be feeding forward to VMN, given that VPN fires before VMN during VOCs (*Bass and Baker, 1990*; *Chagnaud et al., 2011*; *Chagnaud and Bass, 2014*). Consequently, excitatory VPN neurons could activate VPN's glycinergic neurons, via elecrotonic coupling, before the motoneurons (*Figure 9*). This condition is far more likely considering the temporal appearance of the HYP during vocal behavior compared to antidromic (unnatural) conditions. Electrotonic coupling of VPN excitatory neurons to glycinergic VPN neurons thus most likely activates the latter, which causes feed-forward inhibitory action upon motoneurons. Electrophysiological recordings from VPN glycinergic neurons during vocal activity are, however, needed to verify this hypothesis. If proven, we expect these neurons to fire rhythmically like other VPN neurons directly coupled to VMN (*Chagnaud et al., 2011*; also see *Figure 8a*). Anatomical evidence for local activation within VPN comes from transneuronal labeling of local VPN neurons after filling an intracellularly-recorded VPN neuron with neurobiotin (*Chagnaud et al., 2011*; *Chagnaud and Bass, 2014*). While optogenetic inactivation of VPN could provide evidence for this anatomical substrate in the antidromic activation, optogenetic tools are not yet available in toadfishes. Due to the close proximity of VPN to VMN, and its bilateral organization into extended columns along the length of VMN, pharmacological or surgical approaches appear impossible in VPN. Due to the difficulties in obtaining VPN recordings, dual intracellular recordings from VMN and VPN neurons have proven very difficult in the intact preparation (B. Chagnaud, personal observations).

How might antidromic nerve stimulation result in glycine release within such a short time window of a few milliseconds to create HYP? This can only be via electrotonic coupling of glycinergic VPN neurons to the vocal network. Activation of motoneuron axons by antidromic stimulation results in motoneuron depolarization as the AP invades the soma and dendrites. Due to synchronous axon activation, the potential of a substantial number of motoneurons is changed (at high stimulation amplitudes). The potential change in motoneurons is conducted to VPN premotoneurons via electrotonic junctions as premotoneurons are smaller and much more easily excited (*Chagnaud et al., 2011*) than motoneurons. This, in turn, would activate other VPN neurons, including glycinergic ones electrically coupled to VMN and VPN neurons via gap junctions. While transneuronal labeling supports glycinergic VPN neurons being coupled to the VMN network (*Rosner et al., 2018*), we do not know if they contact motoneurons or premotoneurons via gap junctions. Unfortunately, we have so far been unsuccessful using electron microscopy together with the available antibodies for dual labeling of glycinergic and gap junction synapses. In any case, glycinergic neurons are activated and project via a chemical synapse onto motoneurons.

Inhibitory action in this network can serve multiple functions. First, it repolarizes motoneurons, a feature essential to VMN's ability to repetitively fire APs, as seen by our pulse train experiments. Second, it generates a window of decreased excitability that prevents motoneuron misfiring outside of the VPN rhythm due to the long time course of inhibitory action compared to the normal activation rhythm of ca. 100 Hz (see *Video 2*). This is essential to the ability of the next incoming excitatory input of rhythmically firing VPN neurons to set the firing frequency of vocal motoneurons. A gap junction-mediated, feed-forward glycinergic inhibition could account for temporal patterning in the millisecond time range that characterizes the vocal network and behavior of toadfishes (*Pappas and Bennett, 1966*; *Bass and Baker, 1991*; *Chagnaud et al., 2011*; *Chagnaud et al., 2012*). A similar mechanism has previously been shown for the Mauthner cell escape circuit (*Furukawa and Furshpan, 1963*; *Diamond and Roper, 1973*; *Diamond et al., 1973*; *Faber et al., 1989*; *Zottoli and Faber, 2000*; *Korn and Faber, 2005*; *Sillar, 2009*) and may operate in other vocal systems in teleosts and tetrapods that are dependent on comparable levels of temporal precision (*Sturdy et al.,*

*2003*; *Rome, 2006*; *Bass et al., 2015*; *Mead et al., 2017*; *Nelson et al., 2018*; *Kwong-Brown et al., 2019*).

## Consequence of motor–premotor coupling

The notion that vertebrate motoneurons are only passive components has recently changed. Motor–premotor coupling in other systems (*Song et al., 2016*; *Falgairolle et al., 2017*; *Lawton et al., 2017*; *Matsunaga et al., 2017*; *Barkan and Zornik, 2019*) is in line with our observation that motoneuron activity has substantial impact on premotor patterning. While we do not directly show the effect of motoneuronal activity back-propagating through gap junctions on the VPN firing pattern, we hypothesize that the characteristic 'pacemaker' potential that leads to VPN oscillatory-like activity in midshipman (*Chagnaud et al., 2011*) originates from motor–premotor coupling. This would mean that excitatory potentials from the VMN network are shunted by the presumed feed-forward inhibitory action, while the inhibitory potentials would be allowed to travel through gap junctions to contribute to VPN patterning (e.g., see *Figure 8a*; also see *Chagnaud et al., 2011*). Further recordings from VPN neurons are needed to test the contribution of motoneurons to their pattern generating ability. It is, however, conceivable that the gap junctional coupling between VPN and VMN neurons also affects VPN activity and is another example of how premotor and motor neurons interact to generate motor patterns (*Barkan and Zornik, 2019*).

## Functional significance of high temporal precision for vocal behavior

We provide evidence that temporal precision in motoneuronal output from the toadfish vocal network depends on gap junction-mediated, glycinergic inhibition, resulting in a reduced probability of motoneuron activation manifested as a prominent HYP that guards against motoneuron misfiring. Together with the activation of excitatory VPN neurons, the phasic inhibition provides an essential mechanism for generating rapid, highly phasic acoustic modulations determining vocal PRR. Why might there be strong selection for this neurobehavioral mechanism?

From a sensory-motor coupling perspective, teleost auditory systems are exquisitely adapted to temporal coding, including the PRR and fundamental frequency of vocal signals (*Bass and McKibben, 2003b*; *Fay and Edds-Walton, 2008*). As Capranica (*Capranica, 1992*) wrote, the vocal and auditory systems 'co-evolved and we should expect them to share the same underlying code for signal generation and recognition'. From a behavioral perspective, call types differing in temporal properties, including PRR, indicate behavioral state. For example, brief, broadband signals with variable PRRs such as grunts (see *Figure 1a*) inform conspecifics of an aggressive state during defense of resources such as nest sites, whereas mutliharmonic advertisement calls with a more stable PRR (i.e., $F_0$) like the boatwhistle (*Figure 1a*) can function as courtship signals, indicating readiness to mate (for reviews, see *Bass and McKibben, 2003b*; *Ladich et al., 2006*). Underwater playbacks further show that toadfishes, including Gulf toadfish, distinguish call types and PRRs (*Fish, 1972*; *Winn, 1972*; *McKibben and Bass, 1998*; *McKibben and Bass, 2001*; *Remage-Healey and Bass, 2005*). Individual differences in boatwhistle PRR are further linked to male quality and aggressive state in the Lusitanian toadfish, *Halobatrachus didactylus* (*Vasconcelos et al., 2012*; *Amorim et al., 2015b*). More broadly, behavioral evidence supports a role for PRR in individual and species recognition in other soniferous teleosts (*Gerald, 1971*; *Myrberg and Spires, 1972*; *Myrberg and Riggio, 1985*; *Maruska et al., 2007*; *Amorim et al., 2015a*).

As regards sound properties in water, transmission distance is limited by the frequency

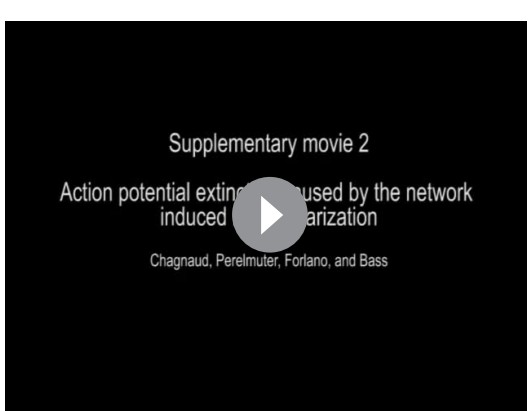

**Video 2.** Intracellular recording of a vocal motoneuron during sequences of repeated antidromic stimulation via the vocal nerve on the ipsilateral and contralateral sides. The latency between the ipsi- and contralateral stimulation was diminished, which eventually abolished the activation of the glycinergic input as motoneurons were hyperpolarized.
https://elifesciences.org/articles/59390#video2

content of calls in shallow water habitats like those where toadfishes and other teleosts defend nests and engage in acoustic courtship (*Gerald, 1971*; *Fine and Lenhardt, 1983*; *Bass and Clark, 2003a*). The principal limitation to call PRR and $F_0$ in species like toadfishes that generate calls with frequency content mainly below 500 Hz is muscle contraction rate that determines swim bladder vibration rate. A superfast motor system, driven by a central network that directly determines high temporal precision in synchronous activation of muscle fibers to maximize call amplitude as well as PRR and $F_0$, enhances transmission distance. While a combination of mechanisms increase temporal precision in the time domain, gap junction-mediated, phasic glycinergic inhibition provides an effective means to achieve temporal precision in motor coding at a network level.

Detection of vocalization fine structure including PRR is also a salient feature of communication in many tetrapods (*Gerhardt et al., 2007*; *Rose, 2014*), including some non-human primates (*Hauser et al., 1998*). Studies of motor coding in tetrapods using nerve or extracellular recordings, or electromyography, also support a role for the hindbrain in determining rapid acoustic events (see *Chagnaud et al., 2011*). We expect mechanisms comparable to those reported here for preventing motoneuron misfiring that, in general, could disrupt distinct transitions in acoustic waveforms to be present in other lineages of vocal vertebrates as well. Even more broadly, they may contribute to time coding in other motor systems requiring high levels of precision.

# Materials and methods

**Key resources table**

| Reagent type (species) or resource | Designation | Source or reference | Identifiers | Additional information |
|---|---|---|---|---|
| Strain, strain background | *Opsanus beta* | Gulf specimens (https://gulfspecimen.org/) | | Specimens taken from the wild |
| Antibody | Anti-rabbit AlexaFluor 568, polyclonal | Thermo Fisher | RRID:AB_143157 | 1:200 |
| Antibody | Anti-rabbit AlexaFluor 568, polyclonal | Thermo Fisher | RRID:AB_2535805 | 1:200 |
| Antibody | Anti-glycine, rabbit, polyclonal | MoBiTec | RRID:AB_2560949 | 1:200 |
| Antibody | Anti-synaptic vesicle protein 2, monoclonal | Developmental Studies Hybridoma Bank | RRID:AB_2315387 | 1:50 |
| Peptide, recombinant protein | 488 anti-streptavidin | Jackson ImmunoResearch Labs | RRID:AB_2337249 | 1:500 |
| Software, algorithm | IGOR PRO | Wavemetrics | RRID:SCR_000325 | |
| Software, algorithm | pClamp | Molecular devices | RRID:SCR_011323 | |
| Software, algorithm | NeuroMatic | http://www.neuromatic.thinkrandom.com | RRID:SCR_004186 | |

## Animals

Gulf toadfish of both sexes (n = 18; six females, standard length: 11.7–13.6 cm; 12 males, 15.3–25.7 cm) were obtained from a commercial source (Gulf Specimen, Panacea, FL) and housed in saltwater aquaria in an environmental control room held at 22℃ on a 14:10 hr light:dark cycle. Though field studies mainly characterize male vocalization, both sexes are capable of acoustic signaling (e.g., *Demski and Gerald, 1972*; *Remage-Healey and Bass, 2005*; *Fine and Thorson, 2008*). Both sexes produce grunts, but only males are known to produce boatwhistles (*Winn, 1967*; *Winn, 1972*; *Thorson and Fine, 2002*). All experimental methods were approved by the Cornell University Institutional Animal Care and Use Committee (#1985-061).

## Surgery for neurophysiology

Surgical and recording methods (see below) were adopted from prior studies (*Bass and Baker, 1990*; *Chagnaud et al., 2012*). Animals were deeply anesthetized during all surgical procedures (immersion in aquarium water containing 0.025% benzocaine [ethyl p-amino benzoate]; Sigma, St.

Louis, MO). For antidromic stimulation of VMN motoneurons, bipolar silver wire electrodes insulated with enamel except at the tips (0.15 mm diameter; 0.3 mm between tips) were implanted at the level of the swim bladder between each muscle and the bladder wall, immediately adjacent to the vocal nerve. A dorsal craniotomy was performed to expose the brainstem and the paired occipital nerves that give rise to the vocal nerve (*Figure 1e*, *Chagnaud and Bass, 2014*). After surgery, animals received an intramuscular injection of bupivacaine anesthetic (0.25%; Abbott Laboratories, Chicago, IL) with 0.01 mg/ml epinephrine (International Medication Systems, El Monte, CA) near the wound site, and then an intramuscular trunk injection of the muscle relaxant pancuronium bromide (0.1–1 µg/g of body weight); bupivacaine was administered every four hours until euthanasia. Animals were placed in a plexiglass tank and perfused over the gills with artificial seawater at 18–20°C.

## Monitoring and activation of vocal motor behavior

Teflon-coated, silver wire electrodes (75 µm diameter) with exposed ball tips (50–100 µm diameter) were used to record the vocal motor volley, hereafter referred to as a fictive vocalization (VOC), from the vocal nerves that innervate the vocal muscles attached to the swim bladder. Signals were amplified 1000-fold and band-pass filtered (300–5000 Hz) with a differential AC amplifier (Model 1700, A-M Systems). VOCs were evoked by current pulses delivered to vocal midbrain areas via insulated tungsten electrodes (5 MΩ impedance; A-M Systems). For display purposes, electrical artifacts were truncated in the illustrations (marked by black arrowheads in *Figures 1*, *2*, *5,* and *6*). Current pulses were delivered via a constant current source (model 305-B, World Precision Instruments). A stimulus generator (A310 Accupulser, World Precision Instruments) was used to generate TTL pulses with a standard stimulus of five pulses at 200 Hz. Each pulse train equaled one stimulus delivery with inter-stimulus intervals of 1 s. During recordings, inter-pulse intervals (100–300 Hz) and total pulse number (2–10) varied. Occasionally, VOCs also occurred spontaneously.

## Neurophysiological recordings

Glass micropipettes (A-M Systems) for intracellular recordings were pulled on a horizontal puller (P97, Sutter Instruments) and were filled with either a 5% neurobiotin solution in 0.5 M KCOOH (resistance 35–50 MΩ) or with 2 M KCOOH. Neuronal signals were amplified 100-fold (Biomedical Engineering) and digitized at a rate of 20 kHz (Digidata 1322A, Axon Instruments/Molecular Devices) using pCLAMP 9 software (Axon Instruments). An external clock (Biomedical Engineering) sending TTL pulses was used to synchronize stimulus delivery and data acquisition. Electrode resistance was monitored while searching for neurons by a current step applied to the recording electrode. In some cases, small amounts of negative current were applied to stabilize the membrane potential after penetration.

## Pharmacological manipulations

The dorsal roots at the level of the vocal occipital roots were cut with iridectomy scissors in two animals. We first recorded the activity of several motoneurons before and several after the transsection. To quantify the contribution of voltage-dependent sodium channels to single motoneuron activity, QX 314-containing electrodes (100 mM in 2 M KCOOH) were used to impale motoneurons and QX314 was electrically driven into the neurons via current injections. The firing patterns directly after the impaling of the neuron and after QX314 injection were compared in the same neurons.

To block gap junction coupling, a CBX solution (10 mM in 0.1 M PB) was injected into the VMN with micropipettes (tip diameters, 20–30 µm). Due to the long time course of gap junction blockage, CBX was also superfused over the fourth ventricle that lies directly above the VMN (see *Figure 1c*, *Rozental et al., 2001*; *Beaumont and Maccaferri, 2011*). After waiting 30–40 min for the CBX to take effect, recordings were resumed.

Strychnine (10 mM in 0.1 M PB) was applied as described above for CBX to block glycine receptors, respectively. After baseline recordings, the pipette solution was pressure-ejected using a picospritzer (Biomedical) set to deliver three pulses, 10–50 ms duration each, at 25–30 PSI at three locations along the rostro-caudal axis of the VMN. Baseline and post-injection vocal activity was recorded from five motoneurons prior to and after strychnine injections in multiple animals.

Lastly, 3 M KCl-filled electrodes were used to increase intracellular chloride concentration in order to reveal inhibitory input. After electrode penetration and a brief recording of baseline activity,

current injection was used to drive chloride ions into the recorded motoneuron, after which its activity was further recorded.

## Data analysis

Neuronal data were processed using Igor pro 6 (Wavemetrics), the software package neuromatic (http://www.neuromatic.thinkrandom.com) with custom written scripts (*Chagnaud, 2020*). The firing pattern of VMN neurons was visualized using a phase plane plot of the recorded voltage (*V*) against the difference in voltage over time (d*V*/d*t*). Summary graphs were generated using the ggplot2 package (*Wickham, 2016*). Violin plots show the distribution of data from all traces, superimposed with the predicted means and standard error from the model. Statistical analysis was performed with R (3.5.1) using the lme4 package (*Bates et al., 2015*) with consultation provided by the Cornell University Statistical Consulting Unit. Data were fit with linear mixed models, treating neuron and animal as random effects, with neuron nested within animal, thus accounting for the nonindependence of multiple recordings of neurons from the same fish. If residual plots revealed that the assumptions of a linear model were not met, data were fit using a generalized linear mixed model. Statistical significance values were determined using the lmerTest package (*Kuznetsova et al., 2017*), fitted using restricted maximum likelihood and the Satterthwaite approximation (*Luke, 2017*). A p-value of less than 0.05 was considered significant. Effect sizes, reported as Cohen's d, were calculated using the EMAtools package. AP and HYP amplitude were measured as follows: the baseline was evaluated by averaging 10 s prior to the stimulus; the maximum and minimum amplitude was then determined during a 30 s window after the stimulation pulse for multiple traces of different neurons. To account for differences in RMP across neurons, we calculated the voltage difference from baseline and included RMP as a covariate. For repeated measures from the same neuron (QX314 and KCl injections), a random coefficient for the effect of treatment on neuron was included. To quantify the effect of strychnine on the variability of motoneuron ISI, we calculated the coefficient of variation from the interval between the first and second spikes across all traces from a given neuron.

## Vocal motoneuron labeling and immunohistochemistry

Surgical and nerve labeling methods follow those in prior neuroanatomical studies (*Bass et al., 1994*; *Chagnaud and Bass, 2014*). As with neurophysiology (see above), animals were deeply anesthetized during all surgical procedures by immersion in aquarium water containing 0.025% benzocaine. Following ventral exposure of one vocal nerve at the rostral pole of the swim bladder, crystals of 10 kDa dextran-biotin (Molecular Probes, Eugene, OR) were applied to the cut end of the nerve. Two juvenile males were deeply anesthetized by immersion in aquarium water containing 0.025% benzocaine and then transcardially perfused with 4% paraformaldehyde and 0.5% glutaraldehyde in 0.1 M phosphate buffer (PB; all Sigma); survival times were 2 (5.3 cm, standard length) or 4 (10.8 cm) days. Brains were postfixed for 1 hr, and then washed and stored in 0.1 M PB at 4°C. Prior to sectioning, brains were transferred to 30% sucrose in 0.1 M PB overnight at 4°C. Brains were embedded in Tissue-Tek O.C.T. compound (Sakura Finetek, Torrence, CA), cryo-sectioned at 25 μm in the transverse plane, and collected onto Superfrost Plus slides (Thermo Fisher Scientific, Waltham, MA). Slides were stored at −80°C.

Slides were rehydrated in 0.1 M PB-saline (PBS, 2 × 15 min). To reduce background autofluorescence from glutaraldehyde, slides were incubated in freshly prepared 0.1% sodium borohydride in PBS for 10 min, followed by washes in PBS (3 × 5 min). Slides were blocked in 10% normal goat serum with 0.5% Triton 100 in PBS (PBS-NGS-T) for 2 hr, then incubated overnight (18 hr) with the following antibodies: anti-glycine (1:200, rabbit, polyclonal, MoBiTec, 1015GE, Göttingen, Germany; see *Rosner et al., 2018* for details on specificity and prior use in other vertebrates, including fish; RRID:AB_2560949) and anti-synaptic vesicle protein 2 (SV2, 1:50, mouse, monoclonal, Developmental Studies Hybridoma Bank, Iowa City, IA; RRID:AB_2315387). The SV2 antibody labels a transmembrane transporter in synaptic vesicles, recognizes all three known isoforms, and demonstrates specificity in both mammals and non-mammals (*Buckley and Kelly, 1985*, manufacturer's information). It labels synaptic vesicles in numerous teleost species (e.g., *Buckley and Kelly, 1985*; *Schikorski et al., 1994*; *Smith et al., 2000*; *Stil and Drapeau, 2016*). Slides were washed in PBS with 0.5% Triton 100 (PBS-T, 4 × 5 min), incubated 4 hr with goat anti-rabbit AlexaFluor 568 (1:200; RRID:AB_143157), goat anti-mouse AlexaFluor 647 (1:200; RRID:AB_2535805), and 488 anti-

streptavidin (1:500; RRID:AB_2337249) in PBS-NGS-T, washed in PBS-T (4 × 5 min), and cover-slipped using Vectashield (Vector Labs, Burlingame, CA) and 1.5H coverglass (Thorlabs, Newton, NJ).

## Super-resolution microscopy

Multicolor images using SR-SIM were acquired with a Zeiss Elyra S.1 SIM system using a 63×/1.4 oil immersion lens and ZEN 2012 software (Zeiss). Z-stacks (0.1 μm step size, 3 μm thick) were acquired using five grid rotations and five phases at each plane. To maximize signal-to-noise, images were acquired within the first 5 μm from the tissue/coverglass interface. SR-SIM images were processed in ZEN, followed by generation of maximum projections and single optical plane images with ImageJ.

## Acknowledgements

Research support from NIH 1428922 to Cornell University Biotechnology Resource Center (BRC) Imaging Facility (Zeiss Elyra super-resolution microscope, SRM), NSF IOS 1457108 and 1656664 to AHB, and DFG CRC 870/B17 to BPC. We thank Luke Remage-Healey (University of Massachusetts, Amherst) for toadfish boatwhistle recording in *Figure 1*, Aaron Rice (Cornell Lab of Ornithology) for photograph of Gulf toadfish in *Figure 1*, and Becky Williams (Cornell BRC) for SRM guidance. The authors acknowledge the excellent comments from anonymous reviewers on a previous version of the manuscript.

## Additional information

### Funding

| Funder | Grant reference number | Author |
| --- | --- | --- |
| National Science Foundation | IOS 1457108 | Andrew H Bass |
| National Science Foundation | IOS 1656664 | Andrew H Bass |
| Deutsche Forschungsgemeinschaft | CRC870/B17 | Boris P Chagnaud |

The funders had no role in study design, data collection and interpretation, or the decision to submit the work for publication.

### Author contributions

Boris P Chagnaud, Conceptualization, Data curation, Software, Formal analysis, Funding acquisition, Investigation, Visualization, Methodology, Writing - original draft, Writing - review and editing; Jonathan T Perelmuter, Investigation, Visualization, Methodology, Writing - original draft, Writing - review and editing; Paul M Forlano, Methodology, Writing - review and editing; Andrew H Bass, Conceptualization, Formal analysis, Funding acquisition, Visualization, Methodology, Writing - original draft, Project administration, Writing - review and editing

### Author ORCIDs

Boris P Chagnaud https://orcid.org/0000-0001-5939-8541
Jonathan T Perelmuter https://orcid.org/0000-0001-9785-8211
Paul M Forlano https://orcid.org/0000-0003-4258-5708
Andrew H Bass https://orcid.org/0000-0002-0182-6715

### Ethics

Animal experimentation: All of the animals were handled according to approved institutional animal care and use committee (IACUC) at Cornell University (#1985-061).

### Decision letter and Author response

Decision letter https://doi.org/10.7554/eLife.59390.sa1
Author response https://doi.org/10.7554/eLife.59390.sa2

## Additional files

### Supplementary files
- Supplementary file 1. Summary of statistical results.
- Transparent reporting form

### Data availability

All data generated or analysed during this study are freely accessible on the Harvard dataverse file repository.

The following dataset was generated:

| Author(s) | Year | Dataset title | Dataset URL | Database and Identifier |
|---|---|---|---|---|
| Chagnaud BP | 2021 | Replication data for Gap junction mediated glycinergic inhibition ensures precise temporal patterning in vocal behavior | https://doi.org/10.7910/DVN/RZLWHE | dataverse.harvard.edu/, 10.7910/DVN/RZLWHE |

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
