## [Decision Letter]

**Acceptance summary:**

Swim bladder muscles are a model for rapid and precise contraction, like those in the rattlesnake tail and the ring dove syrinx. The authors have recorded from the motor neurons of the toadfish sonic motor nucleus to examine the mechanisms underlying their synchronous activation. They show that pronounced temporal precision in population-level firing depends on gap junction-mediated, glycinergic inhibition that generates a period of reduced probability of motoneuron activation. Super-resolution microscopy reveals glycinergic release sites from nearby pre-motor neurons onto the motor neurons. The authors propose gap junction-mediated, glycinergic inhibition provides a timing mechanism for achieving synchrony and temporal precision in the millisecond range for rapid modulation of this network.

**Decision letter after peer review:**

[Editors’ note: the authors submitted for reconsideration following the decision after peer review. What follows is the decision letter after the first round of review.]

Thank you for submitting your work entitled "Gap junction mediated feed-forward inhibition ensures ultra precise temporal patterning in vocal behavior" for consideration by *eLife*. Your article has been reviewed by three peer reviewers, one of whom is a member of our Board of Reviewing Editors, and the evaluation has been overseen by a Senior Editor. The reviewers have opted to remain anonymous.

Our decision has been reached after consultation between the reviewers. Based on these discussions and the individual reviews below, we regret to inform you that your work will not be considered further for publication in *eLife*.

Reviewer #1:

The authors have replicated the findings of Pappas and Bennett in the toadfish sonic motor nucleus, and then described the role of inhibition of network elements in the generation of an AHP, which appears to allow continued firing. The authors provide support for the hypothesis that the presence of the AHP at high levels of motoneuron recruitment strongly suggests a network-dependent activation of the AHP. There are some concerns that detract from my enthusiasm for an otherwise thorough study. Principally these concerns stem from unsupported statements, but unfortunately, the paper also does not identify the source of the glycinergic neurons.

1) The synchronous firing does not conform to the typical performance for extreme temporal precision in motoneuronal firing, I suggest the authors modify these statements. The authors state "The extent of temporal precision across an entire neuronal population generated in toadfish far exceeds the magnitude of synchronization known for other vocal circuits", but provide no data to support the statement.

2) The authors make a strong case for network-dependent activation of the AHP, via glycine action. First, glycinergic roles in achieving temporal precision are well known (see Grothe references, for example). The network action is interesting, and the authors convincingly demonstrate the presence of glycinergic boutons. Nevertheless, since the role of glycine in AHP is the major finding, it would seem important to identify the source of this input.

3) With respect to the AHP, the authors write that it is essential to generating a highly synchronous motor output...with extreme temporal precision, but do not test this hypothesis.

4) Some data do not appear to contribute to the paper in a significant way. For example, the intracellular injections of chloride to not appear to add to our understanding.

Reviewer #2:

The authors report recordings from swim bladder muscle motoneurons and their inputs to study the capacity of toadfish to rapidly set its swim bladder muscles into sustained and rapid oscillation, a feature that clearly requires a degree of synchrony of firing in the motoneurons. The authors also report evidence suggestive of feed-forward inhibition in this system which would be remarkable in this system. Pharmacological experiments and SR-SIM microscopy on dextran-filled motoneurons support the notion of feedforward-inhibition and narrow down the nature of the chemical synapses involved. In its present stage I do however have a number of major concerns that need to be addressed:

1) Novelty-concerns. It is essential that the authors clarify in how far their measurements and interpretation go beyond those reported in Pappas and Bennett, 1966. This study already reports many of the experiments that are presented here and so it would be important to work out as clearly as possible the specific focus of the present study. This would probably also help to substantially shorten the complex manuscript and to emphasize the specific merits of the new findings.

2) Direct evidence for feedforward inhibition.

All findings nicely fit together and are very clearly compatible with feedforward inhibition but they are not yet conclusive (and the authors are very clear about this point). I imagine that it will be very difficult to try double intracellular recordings and I therefore do not want to suggest experiments that simply cannot be done in this system. But I wonder whether it might be feasible to attempt multiple simultaneous patch-clamp recordings for an additional line of reasoning?

3) “ultra precise temporal patterning”

While it is clear that the activity must cause powerful in-phase contractions of the swim bladder I fail to see why this requires “ultra-precise temporal patterning”. With a jitter of 1.5 ms in a periodic event of 4 ms the coefficient of variation is in the low-accuracy range of periodic biological phenomena. I think the term is more of an obstacle for seeing the beauty of the system. It should either be explained better or avoided.

4) Basic writing, Figures, Data presentation: I found the manuscript exceptionally difficult to read, although I am sure that the authors thought very carefully about what they report. I strongly suggest that all figure legends (including the supplementary figures) should start with a first sentence that says what lesson is to be learned from the particular figure.

My prediction is that the paper would very much profit from shortening and from a focus on the specific effects that are (A) new relative to the pioneering work e.g. of Pappas and Bennett (1966 )and (B) makes the system interesting despite the wonderful and direct evidence on feed-forward inhibition in the particularly accessible Mauthner system. Also, a small schematic of the proposed connections might be helpful to shorten the text and to guide readers of what’s currently examined. Regarding data presentation: All N's need to reported with an indication of how many trials were contributed from each preparation. e.g, 3 fish, 7 cells, with n=2 to 9 recordings each. All asterisks and stats should also be explained in the figure legends as well as details shown in the figures (e.g. scaling/labels in the phase plots etc)

Reviewer #3:

In this manuscript, the authors carried out a series of elegant electrophysiological and anatomical experiments and identified glycine-mediated hyperpolarization after spiking that appear to be important for temporal patterns of the vocalizations produced by the toadfish. By comparing intracellular recordings obtained from the vocal motor neurons during fictive vocalizations, current step injections, and antidromic stimulation, the authors were able to make a great advance in the understanding of how neuronal activity is synchronized. However, there are some issues that I will list below.

1) The authors propose that the glycinergic inputs that mediate HYP originate from the premotor nucleus, which presumably are activated by descending inputs in a feed-forward manner. However, they did not perform a critical experiment to make this conclusion. VPN should be removed (either pharmacologically or surgically) from the vocal circuitry and examine if the action potentials elicited by the antidromic stimulation is followed by HYP. If VPN is the source of glycinergic inputs, HYP is expected to be absent in this preparation. Given that this is one of the major conclusions of the manuscript, the experiment is necessary.

2) In most electrophysiological experiments, the authors repeated experiments multiple times using the same neuron, and treated each data as an independent sample (n=21 from 5 neurons, for example). However, the replicate data are still obtained from the same cell, so they cannot be considered as independent samples. Thus, the issue of pseudoreplication should be rectified.

[Editors’ note: further revisions were suggested prior to acceptance, as described below.]

Thank you for submitting your article "Gap junction mediated glycinergic inhibition ensures precise temporal patterning in vocal behavior" for consideration by *eLife*. Your article has been reviewed by two peer reviewers, including Catherine Emily Carr as the Reviewing Editor and Reviewer #1, and the evaluation has been overseen by Ronald Calabrese as the Senior Editor.

The reviewers have discussed their reviews with one another, and the Reviewing Editor has drafted this to help you prepare a revised submission. The paper is ready to be accepted except for one reviewer made some useful comments; the paper will not go out for review again, but we thought you would prefer to make the corrections and address the question about QX314.

Summary:

Swim bladder muscles are a model for rapid and precise contraction, like those in the rattlesnake tail and the ring dove syrinx. The authors have recorded from the motor neurons of the toadfish sonic motor nucleus to examine the mechanisms underlying their synchronous activation. They show that pronounced temporal precision in population-level firing depends on gap junction-mediated, glycinergic inhibition that generates a period of reduced probability of motoneuron activation. Super-resolution microscopy reveals glycinergic release sites from nearby pre-motor neurons onto the motor neurons. The authors propose gap junction-mediated, glycinergic inhibition provides a timing mechanism for achieving synchrony and temporal precision in the millisecond range for rapid modulation of this network.

Reviewer #2:

The authors have done a magnificent job in reorganizing and rewritten their manuscript and I would like to thank them for taking my concerns and those of my colleagues so seriously. I think it was really worth the effort. The importance of the work, its major thrust, and its underlying logic and importance have become clear now as has its building upon the findings of Pappas and Bennet, its novelty, its neuroethological importance, and potential significance on a wider scope. I enjoyed reading the new manuscript and also seeing its remarkable transformation. I have just very few remaining points that should be addressed but do not require formal revision.

1) The interpretation of the experiments with intracellularly injected QX314 is convincing but should be complemented with something to convince readers that you would have gotten sufficient QX314 into the neurons and (if possible) to say that QX314 also acts on the sodium channels in the acoustic motoneurons you used. I am asking, because a positive control (with an isolated motoneuron) is impossible. Perhaps positive controls from a test injection into a similar sized neuron, when available, would help. In the event that nothing could be said on this point, then it would be important to add that, even though unlikely, QX314 may not have been injected in insufficient quantity.

2) Subsection “Dependence of HYP activation on glycinergic inhibitory input to VMN” paragraph two. 400 ms: is this really correct? Why would this be a “short” delay? If not a typo I'd be lost here. And: p23 l712 were any controls made to say that the injection would have gotten enough substance into the neuron?

---

## [Author Response]

[Editors’ note: The authors appealed the original decision. What follows is the authors’ response to the first round of review.]

Reviewer #1:The authors have replicated the findings of Pappas and Bennett in the toadfish sonic motor nucleus, and then described the role of inhibition of network elements in the generation of an AHP, which appears to allow continued firing. The authors provide support for the hypothesis that the presence of the AHP at high levels of motoneuron recruitment strongly suggests a network-dependent activation of the AHP.

In regards to novelty (also raised by reviewer 2, see below), we cited the pioneering work of *Pappas and Bennett* throughout the original submission. By no means was it our intention not to give appropriate credit to Pappas and Bennett and if it came across that way then it was sloppy writing by us. We realize in hindsight that we should have been more direct about what we report compared to Pappas and Bennett (1966). We now do this in paragraphs six to eight of the Introduction and in sections of the Results (Vocal premotor neurons are excited by gap junctional coupling; Motoneuron electrical coupling and HYP [hyperpolarization], Vocal premotor neurons are excited by gap junctional coupling) and Discussion (Electrical coupling and inhibition in vocal network). We also more clearly state in paragraph six of the Introduction that Pappas and Bennett only speculated on the role of inhibition in output of the vocal (sonic) motor nucleus (VMN) without experimentally testing it. Unlike Pappas and Bennett, we provide evidence for glycinergic inhibition dependent on gap junctional coupling in this system. This rests upon: (1) evidence from gap junctional coupling of the inhibition (Results section: Motoneuron electrical coupling and HYP), (2) blocking gap junctions (Results: Pharmacological manipulations), (3) testing for glycinergic dependency of HYP (Results: Glycine-dependent HYP), (4) recording from motoneuron afferents whose identity was unknown at the time to show that they are VPN neurons, and that they could be activated by antidromic stimulation of motoneurons (the latter had been shown in part by Pappas and Bennett; Results: Vocal premotor neurons excited by gap junctional coupling). Paragraph four of the revised Introduction also more clearly explains that individual vocal nerve potentials in this neurophysiological preparation represent the synchronous firing of motoneurons. Perhaps equally important, the nerve potentials comprising an entire motor volley (*fictive vocalization*) determine the physical attributes of natural vocal behaviors.

There are some concerns that detract from my enthusiasm for an otherwise thorough study. Principally these concerns stem from unsupported statements, but unfortunately, the paper also does not identify the source of the glycinergic neurons.

The need to better substantiate “unsupported statements” (also see response below to reviewer 2) largely arose from misinterpretations due to our writing style (this was clearly our fault), e.g., what we meant by synchrony (see below). We agree that it was critical to “identify the source of the glycinergic neurons“. As we stated in the original manuscript, “We previously identified a subpopulation of glycinergic VPN neurons, suggesting direct glycinergic input onto motoneurons (Rosner et al., 2018)”. Rosner et al. is a publication in the Journal of Comparative Neurology showing the location of transneuronally labeled (gap junction coupled) glycinergic somata within the column of vocal pacemaker neurons (We meant the delay between midbrain stimulation and antidromic stimulation). We deleted this statement as it added confusion.

That are adjacent to, and directly innervate, the VMN (see paragraph four of revised manuscript). This anatomical finding was one of the major reasons that we pursued the key glycinergic experiments reported in our manuscript. We have now revised Figure 1 to illustrate this background information (Figure 1F). The super resolution imaging in the current paper was intended to only show glycinergic synaptic boutons located on motoneuron somata, and whose presence we only speculated on in the earlier Journal of Comparative Neurology paper. As we also report (next to last section of Results), we recorded from VPN neurons to show that they can be electrotonically activated.

We have also revised the title of the paper in two major ways. We removed “ultra” from the title and from the text in response to the above and comments by reviewers 2 and 3 (see below). We also removed feed-forward from the title as this was too strong a claim – we now consider both feed-forward and feedback in the Discussion – Feed-forward or feedback inhibition?

In response to other major comments by reviewers and as discussed more below, we also changed the title of the manuscript and shortened and streamlined the manuscript in several ways.

1) The synchronous firing does not conform to the typical performance for extreme temporal precision in motoneuronal firing, I suggest the authors modify these statements. The authors state "The extent of temporal precision across an entire neuronal population generated in toadfish far exceeds the magnitude of synchronization known for other vocal circuits", but provide no data to support the statement.

Here is an example of where our writing appeared to mislead the reviewers to equate *repetition rate* with *sychnrony*. We recognize that we needed to do a much better job of clarifying this important point. This is now more completely explained in paragraph four of the Introduction. To test the contribution of single motoneurons to vocal network activity, we performed intracellular recordings of motoneurons using QX314 that blocks voltage-dependent sodium channels intracellularly (Figure 5). As stated in the Results, “These experiments demonstrated that (i) the contribution of the recorded motoneuronal AP firing to the firing of that motoneuron is rather small during VOC activity (the activity of the neuron is dominated by gap junction coupled potentials), (ii) during a VOC, the HYP amplitude of the recorded neuron only partly depends on its firing an AP, and (iii) most of the activity displayed by a given motoneuron reflects population-level motoneuronal activity.”

2) The authors make a strong case for network-dependent activation of the AHP, via glycine action. First, glycinergic roles in achieving temporal precision are well known (see Grothe references, for example). The network action is interesting, and the authors convincingly demonstrate the presence of glycinergic boutons. Nevertheless, since the role of glycine in AHP is the major finding, it would seem important to identify the source of this input.

We appreciate the reviewer’s recognition that we **“**make a strong case for network-dependent activation of the AHP, via glycine action”. We cited Benedikt Grothe’s work already in the initial submission, although it is in a different context than ours: sensory vs motor (paragraph one, Introduction). As stated above, we described the cells of origin of the glycinergic input in Rosner et al., 2018.

3) With respect to the AHP, the authors write that it is essential to generating a highly synchronous motor output…with extreme temporal precision, but do not test this hypothesis.

This is an example of a flaw in our writing that likely misled the reviewers as to what we meant by *synchronous motor output* and thus also appeared to be an *unsupported statement* (see above). As stated above, the QX314 experiment showed that during a VOC, the recorded neuron only partly contributes to the recorded response; it more completely depends on synchronous population-level motoneuron activity. The influence of the HYP on synchronous motor output was tested by injecting strychnine into the vocal motor nucleus (VMN). Immediately after strychnine injection, the HYP of VMN motoneurons disappeared. Similarly, the HYP during antidromic stimulation was completely abolished, symptomatic of decreased synchrony within the VMN population. After strychnine injections, spontaneous VOCs appeared with similar decreases in motoneuron HYP amplitude not seen for spontaneous VOCs under control conditions without prior strychnine injections.

4) Some data do not appear to contribute to the paper in a significant way. For example, the intracellular injections of chloride to not appear to add to our understanding.

We understand the reviewer comment and modified the text. We believe, however, that a paper dealing with inhibition that does not change the chloride reversal potential of neurons is not complete. We moved the panel showing the effects of intracellular chloride injection to the new Figure 7—figure supplement 1.

Reviewer #2:The authors report recordings from swim bladder muscle motoneurons and their inputs to study the capacity of toadfish to rapidly set its swim bladder muscles into sustained and rapid oscillation, a feature that clearly requires a degree of synchrony of firing in the motoneurons. The authors also report evidence suggestive of feed-forward inhibition in this system which would be remarkable in this system. Pharmacological experiments and SR-SIM microscopy on dextran-filled motoneurons support the notion of feedforward-inhibition and narrow down the nature of the chemical synapses involved. In its present stage I do however have a number of major concerns that need to be addressed:1) Novelty-concerns. It is essential that the authors clarify in how far their measurements and interpretation go beyond those reported in Pappas and Bennett 1966. This study already reports many of the experiments that are presented here and so it would be important to work out as clearly as possible the specific focus of the present study. This would probably also help to substantially shorten the complex manuscript and to emphasize the specific merits of the new findings.

Please see response above to reviewer 1’s opening paragraph.

2) Direct evidence for feedforward inhibition.All findings nicely fit together and are very clearly compatible with feedforward inhibition but they are not yet conclusive (and the authors are very clear about this point). I imagine that it will be very difficult to try double intracellular recordings and I therefore do not want to suggest experiments that simply cannot be done in this system. But I wonder whether it might be feasible to attempt multiple simultaneous patch-clamp recordings for an additional line of reasoning?

Although we did not perform patch clamp recordings, we recorded successively from motoneurons and premotoneurons, which we considered a major accomplishment in this in vivo preparation with no visual guidance to the respective neurons. We can confirm from experience that it is, “very difficult to try double intracellular recordings“. The suggested experiment has been done by the El Manira lab, but in genetically modified zebrafish (under visual guidance), although they did not show that the inhibitory neurons were activated above their action potential threshold (see Song et al., 2016, reference in our manuscript) as we were able to show.

3) “ultra precise temporal patterning”While it is clear that the activity must cause powerful in-phase contractions of the swim bladder I fail to see why this requires “ultra-precise temporal patterning”. With a jitter of 1.5 ms in a periodic event of 4 ms the coefficient of variation is in the low-accuracy range of periodic biological phenomena. I think the term is more of an obstacle for seeing the beauty of the system. It should either be explained better or avoided.

We agree that we needed to better explain what we meant by *“*ultra-precise temporal patterning*”*. In the original manuscript, we did not clearly distinguish properties of the motor volley, the fictive vocalization (*VOC*), from synchrony. We hope to have now done a much better job of making this distinction in the latter half of paragraph four of the Introduction (sentence beginning “The motor volley’s highly stereotyped, repetitive series…”). When an entire nucleus with hundreds of neurons fires within 1.5 ms, that is synchronous. As we were referring to individual VOC potentials, the coefficient of variance is in this instance not applicable. We have removed the word “ultra” from the title and from the manuscript.

4) Basic writing, Figures, Data presentation: I found the manuscript exceptionally difficult to read, although I am sure that the authors thought very carefully about what they report. I strongly suggest that all figure legends (including the supplementary figures) should start with a first sentence that says what lesson is to be learned from the particular figure. My prediction is that the paper would very much profit from shortening and from a focus on the specific effects that are (A) new relative to the pioneering work e.g. of Pappas and Bennett (1966 )and (B) makes the system interesting despite the wonderful and direct evidence on feed-forward inhibition in the particularly accessible Mauthner system.

We have extensively modified the manuscript taking all of the reviewers comments into account and hope that it reads much better now. All figure legends (including the supplementary figures) start with a first sentence that says what lesson is to be learned from the particular figure. We also shortened and streamlined the manuscript in several ways. The background information in the original section 1 of the Results has been incorporated into the Introduction. The section titled “Dependence of HYP activation on inhibitory input to VMN” now focuses on our new findings regarding glycinergic input to VMN, a major focus of this report. This section first presents anatomical results, originally a separate section at the end of the Results, showing glycinergic contacts on vocal motoneuron somata and dendrites. We then focus on glycine-related neurophysiology. We removed the results showing effect of bicuculline injection as these were redundant with prior ones reported for the closely related midshipman fish. As noted above, we also moved the panel showing the effects of intracellular chloride injection to Figure 7—figure supplement 1; this also now includes the results showing that sub-threshold midbrain stimulation induces tonic membrane hyperpolarizations.

We refer to the Mauthner system in the Discussion, even though this is a totally different condition: repetitive inhibition for pattern generation vs single “one shot” inhibition event in the Mauthner system.

Also, a small schematic of the proposed connections might be helpful to shorten the text and to guide readers of what’s currently examined. Regarding data presentation: All N's need to reported with an indication of how many trials were contributed from each preparation. e.g, 3 fish, 7 cells, with n=2 to 9 recordings each. All asterisks and stats should also be explained in the figure legends as well as details shown in the figures (e.g. scaling/labels in the phase plots etc)

We agree with the reviewer on all of these points. We now include a summary figure 9. All asterisks and stats should also be explained in the figure legends. All N's are reported with an indication of how many trials were contributed from each preparation. We also added scaling labels to the phase plane plots (we initially omitted these as the PPP’s were intended to show differences between the conditions and not absolute values).

Reviewer #3:In this manuscript, the authors carried out a series of elegant electrophysiological and anatomical experiments and identified glycine-mediated hyperpolarization after spiking that appear to be important for temporal patterns of the vocalizations produced by the toadfish. By comparing intracellular recordings obtained from the vocal motor neurons during fictive vocalizations, current step injections, and antidromic stimulation, the authors were able to make a great advance in the understanding of how neuronal activity is synchronized. However, there are some issues that I will list below.1) The authors propose that the glycinergic inputs that mediate HYP originate from the premotor nucleus, which presumably are activated by descending inputs in a feed-forward manner. However, they did not perform a critical experiment to make this conclusion. VPN should be removed (either pharmacologically or surgically) from the vocal circuitry and examine if the action potentials elicited by the antidromic stimulation is followed by HYP. If VPN is the source of glycinergic inputs, HYP is expected to be absent in this preparation. Given that this is one of the major conclusions of the manuscript, the experiment is necessary.

It would be great to remove VPN (either pharmacologically or surgically), from the vocal circuitry and examine if the action potentials elicited by the antidromic stimulation is followed by a HYP. However, like the difficulty of double intracellular recordings in this intact, *in situ* preparation (Reviewer 2), the organization of the VPN as a long extended column directly opposed to the VMN makes this an impossible task in this system (but doable with optogenetics in a genetically modified system like zebrafish). We included the reviewer’s comment into the discussion.

2) In most electrophysiological experiments, the authors repeated experiments multiple times using the same neuron, and treated each data as an independent sample (n=21 from 5 neurons, for example). However, the replicate data are still obtained from the same cell, so they cannot be considered as independent samples. Thus, the issue of pseudoreplication should be rectified.

We have revised our analyses and illustrations to address this important issue.To account for the non-independence of multiple recordings of neurons from the same animal, we reanalyzed our data using linear and generalized linear mixed models. This approach accommodates nested data structures, avoiding issues of pseudoreplication. This statistical correction only changed the following original result that had no consequences for the overall interpretation of the results. We originally reported that chloride injections reduced the antidromically evoked hyperpolarization. In the new analysis, this difference is no longer statistically significant (p = 0.093).

We originally only looked at intervals between VOC potentials after strychnine application. We now also look at the intervals between motoneuron action potentials, showing significant increases in the frequency of the interspike interval as well as in the variability of the intervals, as measured by the coefficient of variation.

[Editors’ note: what follows is the authors’ response to the second round of review.]

Reviewer #2:The authors have done a magnificent job in reorganizing and rewritten their manuscript and I would like to thank them for taking my concerns and those of my colleagues so seriously. I think it was really worth the effort. The importance of the work, its major thrust, and its underlying logic and importance have become clear now as has its building upon the findings of Pappas and Bennet, its novelty, its neuroethological importance, and potential significance on a wider scope. I enjoyed reading the new manuscript and also seeing its remarkable transformation. I have just very few remaining points that should be addressed but do not require formal revision.1) The interpretation of the experiments with intracellularly injected QX314 is convincing but should be complemented with something to convince readers that you would have gotten sufficient QX314 into the neurons and (if possible) to say that QX314 also acts on the sodium channels in the acoustic motoneurons you used. I am asking, because a positive control (with an isolated motoneuron) is impossible. Perhaps positive controls from a test injection into a similar sized neuron, when available, would help. In the event that nothing could be said on this point, then it would be important to add that, even though unlikely, QX314 may not have been injected in insufficient quantity.

At this point it is not possible to do control injections, as the first author [who has done the physiological experiments; BPC] is on the other side of the Atlantic Ocean (Austria), yet the fishes are only available in the US. As requested by the reviewer we have added a statement into the text: “Even though unlikely, due to the effect on the antidromic AP, an alternative interpretation is that an insufficient quantity of QX 314 was injected.”

2) Subsection “Dependence of HYP activation on glycinergic inhibitory input to VMN” paragraph two. 400 ms: is this really correct? Why would this be a “short” delay? If not a typo I'd be lost here. And: p23 l712 were any controls made to say that the injection would have gotten enough substance into the neuron?

We meant the delay between midbrain stimulation and antidromic stimulation. We deleted this statement as it added confusion.